# Channel nuclear pore complex subunits are required for transposon silencing in *Drosophila*

**Marzia Munafò[1], Victoria R Lawless[1], Alessandro Passera[1], Serena MacMillan[1], Susanne Bornelöv[1], Irmgard U Haussmann[2,3], Matthias Soller[3,4], Gregory J Hannon[1]\*, Benjamin Czech[1]\***

[1]Cancer Research UK Cambridge Institute, University of Cambridge, Li Ka Shing Centre, Cambridge, United Kingdom; [2]Department of Life Science, Faculty of Health, Education and Life Sciences, Birmingham City University, Birmingham, United Kingdom; [3]School of Biosciences, College of Life and Environmental Sciences, University of Birmingham, Birmingham, United Kingdom; [4]Birmingham Center for Genome Biology, University of Birmingham, Birmingham, United Kingdom

**Abstract** The nuclear pore complex (NPC) is the principal gateway between nucleus and cytoplasm that enables exchange of macromolecular cargo. Composed of multiple copies of ~30 different nucleoporins (Nups), the NPC acts as a selective portal, interacting with factors which individually license passage of specific cargo classes. Here we show that two Nups of the inner channel, Nup54 and Nup58, are essential for transposon silencing via the PIWI-interacting RNA (piRNA) pathway in the *Drosophila* ovary. In ovarian follicle cells, loss of Nup54 and Nup58 results in compromised piRNA biogenesis exclusively from the *flamenco* locus, whereas knockdowns of other NPC subunits have widespread consequences. This provides evidence that some Nups can acquire specialised roles in tissue-specific contexts. Our findings consolidate the idea that the NPC has functions beyond simply constituting a barrier to nuclear/cytoplasmic exchange as genomic loci subjected to strong selective pressure can exploit NPC subunits to facilitate their expression.

**\*For correspondence:**
greg.hannon@cruk.cam.ac.uk (GJH);
benjamin.czech@cruk.cam.ac.uk (BC)

**Competing interests:** The authors declare that no competing interests exist.

## Introduction

The main gateway between the nucleus and the cytoplasm is the nuclear pore complex (NPC), a large multi-protein assembly spanning the nuclear envelope. The NPC is composed of multiple copies of ~30 proteins, termed nucleoporins (Nups), arranged into an eightfold symmetric ring (*Beck and Hurt, 2017*; *Hampoelz et al., 2019*; *Kim et al., 2018*). Small molecules can freely diffuse across the NPC, whilst particles larger than 40 kDa or 5 nm require active transport. Transcripts that have passed nuclear quality control steps are actively trafficked across the NPC towards their target sites (*Tutucci and Stutz, 2011*) and dedicated protein networks ensure that transcripts going through the NPC reach their correct cytoplasmic destinations (*Köhler and Hurt, 2007*; *Tutucci and Stutz, 2011*). The NPC has been implicated as more than a simple gateway, serving also as an active player in gene regulation (*Köhler and Hurt, 2010*; *Strambio-De-Castillia et al., 2010*). Some Nups associate with chromatin, displaying preferences for certain epigenetic modifications (*Capelson et al., 2010*; *Gozalo et al., 2020*; *Iglesias et al., 2020*; *Kalverda et al., 2010*; *Vaquerizas et al., 2010*), inducible genes sometimes re-locate proximally to the NPC upon activation (*Blobel, 1985*; *Dieppois et al., 2006*; *Luthra et al., 2007*; *Rohner et al., 2013*; *Strambio-De-Castillia et al., 2010*), and other Nups contribute to heterochromatin organisation and epigenetic inheritance (*Holla et al., 2020*; *Iglesias et al., 2020*). Notably, altered expression or mutation of

**eLife digest** Transposons are genetic sequences, which, when active, can move around the genome and insert themselves into new locations. This can potentially disrupt the information required for cells to work properly: in reproductive organs, for example, transposon activity can lead to infertility. Many organisms therefore have cellular systems that keep transposons in check.

Animal cells comprise two main compartments: the nucleus, which contains the genetic information, and the cytosol, where most chemical reactions necessary for life take place. Molecules continually move between nucleus and cytosol, much as people go in and out of a busy train station. The connecting 'doors' between the two compartments are called Nuclear Pore Complexes (NPCs), and their job is to ensure that each molecule passing through reaches its correct destination.

Recent research shows that the individual proteins making up NPCs (called nucleoporins) may play other roles within the cell. In particular, genetic studies in fruit flies suggested that some nucleoporins help to control transposon activity within the ovary – but how they did this was still unclear. Munafò et al. therefore set out to determine if the nucleoporins were indeed actively silencing the transposons, or if this was just a side effect of altered nuclear-cytosolic transport.

Experiments using cells grown from fruit fly ovaries revealed that depleting two specific nucleoporins, Nup54 and Nup58, re-activated transposons with minimal effects on most genes or the overall health of the cells. This suggests that Nup54 and Nup58 play a direct role in transposon silencing.

Further, detailed analysis of gene expression in Nup54- and Nup58-lacking cells revealed that the product of one gene, *flamenco*, was indeed affected. Normally, *flamenco* acts as a 'master switch' to turn off transposons. Without Nup54 and Nup58, the molecule encoded by *flamenco* could not reach its dedicated location in the cytosol, and thus could not carry out its task.

These results show that, far from being mere 'doorkeepers' for the nucleus, nucleoporins play important roles adapted to individual tissues in the body. Further research will help determine if the same is true for other organisms, and if these mechanisms can help understand human diseases.

certain Nups can cause human diseases that only affect specific tissues, despite the NPC being ubiquitous (*Beck and Hurt, 2017*). This suggests that some Nups might have evolved tissue-specific functions, though the nature of these remains elusive.

Transposable element (TE) silencing in animal gonads is accomplished primarily through the action of piRNAs (*Czech et al., 2018*; *Ozata et al., 2019*). These 23- to 30-nt small RNAs guide PIWI-clade Argonaute proteins to recognise and silence active TEs. piRNAs originate from discrete genomic loci, termed piRNA clusters, largely composed of TE remnants (*Aravin et al., 2006*; *Brennecke et al., 2007*; *Mohn et al., 2014*). In *Drosophila melanogaster*, dual-strand clusters produce RNAs from both genomic strands in the nurse cells of the ovary (*Figure 1—figure supplement 1A*) and rely on non-canonical transcription and export mechanisms (*Andersen et al., 2017*; *ElMaghraby et al., 2019*; *Kneuss et al., 2019*; *Mohn et al., 2014*; *Zhang et al., 2014*). Specification of these transcripts for piRNA production takes place in perinuclear structures, namely nuage (*Lim and Kai, 2007*; *Malone et al., 2009*; *Senti et al., 2015*). Uni-strand clusters instead are transcribed from only one genomic strand and appear to be conventional RNA polymerase II transcripts (*Brennecke et al., 2007*; *Dennis et al., 2016*; *Goriaux et al., 2014*; *Mohn et al., 2014*). Of the two *Drosophila* uni-strand clusters, *flamenco* (*flam*) is the principal source of piRNAs in the somatic follicle cells that enclose egg chambers (*Figure 1—figure supplement 1A*; *Brennecke et al., 2007*; *Malone et al., 2009*) and was originally identified as a master regulator of *gypsy* retrotransposons (*Mével-Ninio et al., 2007*; *Pélisson et al., 1994*; *Prud'homme et al., 1995*). Being an unusually large transcriptional unit, *flam* covers up to ~650 kb of pericentromeric heterochromatin of chromosome X and depends on conventional RNA export mechanisms, centred on the nuclear export factor heterodimer Nxf1/Nxt1 (*Dennis et al., 2016*; *Herold et al., 2001*; *Tutucci and Stutz, 2011*). Upon export, *flam* transcripts localise to perinuclear Yb-bodies, where they are thought to be licensed for piRNA biogenesis (*Hirakata et al., 2019*; *Murano et al., 2019*; *Qi et al., 2011*; *Saito et al., 2010*), though the underlying molecular mechanisms are not fully understood.

Yb-bodies are cytoplasmic, perinuclear condensates of the DEAD-box RNA helicase Yb, encoded by the *fs(1)Yb* gene, which is exclusively expressed in somatic follicle cells (*Figure 1—figure supplement 1A*; *Hirakata et al., 2019*; *Olivieri et al., 2010*; *Qi et al., 2011*; *Saito et al., 2010*; *Szakmary et al., 2009*). Yb is essential for somatic piRNA production, and its assembly into Yb-bodies does not depend on any known piRNA biogenesis factor, and therefore is at the apex of the piRNA biogenesis protein network (*Hirakata et al., 2019*; *Ishizu et al., 2015*; *Ishizu et al., 2019*; *Murota et al., 2014*; *Olivieri et al., 2010*; *Saito et al., 2010*). Yb-bodies have been reported to possess biophysical properties typical of phase-separated condensates, which likely facilitate the biochemical processes happening in these foci (*Hirakata et al., 2019*). Given that *flam* is the major piRNA source locus in somatic follicle cells, we and others have found that the formation of Yb-bodies depends on *flam* RNA (*Figure 1—figure supplement 1*; *Dennis et al., 2016*; *Hirakata et al., 2019*; *Sokolova et al., 2019*). Two previously described *flam* mutant alleles (*Brennecke et al., 2007*; *Malone et al., 2009*; *Mével-Ninio et al., 2007*) disrupt piRNA cluster transcription via *P-element* insertions near the 5′ end: *flam^{BG}*, carrying an insertion in the putative promoter, and *flam^{KG}*, carrying an insertion immediately downstream of the TSS (*Figure 1—figure supplement 1B*). The strongest effect on *flam* expression is observed in trans-heterozygous flies (*flam^{BG/KG}*), obtained through crosses between the two alleles. These mutants show strong de-repression of somatic, *gypsy*-family TEs (*Brennecke et al., 2007*; *Malone et al., 2009*; *Mével-Ninio et al., 2007*; *Figure 1—figure supplement 1C, D*), which is accompanied by a disassembly of Yb-bodies, despite some of the protein still being present (*Figure 1—figure supplement 1E–G*). Notably, the production of other classes of piRNAs, such as those derived from the 3′ UTR of coding genes, is unchanged in these mutants (*Hirakata et al., 2019*; *Sokolova et al., 2019*), further underscoring a link between *flam* and Yb-bodies. Knockdowns of *nxf1/nxt1* in ovarian somatic cells (OSCs), a cell line derived from the somatic compartment of the ovary that expresses a functional piRNA pathway (*Saito et al., 2009*), also compromise Yb-bodies formation (*Figure 1—figure supplement 1A,* H), with similar results reported in soma-specific knockdowns in ovaries (*Dennis et al., 2016*; *Sokolova et al., 2019*). These results, together with previous findings (*Dennis et al., 2016*; *Hirakata et al., 2019*; *Sokolova et al., 2019*), suggest that the production and localisation of *flam* transcripts to Yb-bodies and the assembly of those structures are interdependent. Nonetheless, it is still unknown how the transcript is specifically directed to Yb-bodies and, from there, licensed for processing.

Here, by investigating the export and licensing of *flam*, we uncover a requirement of specific channel Nups for TE silencing in the somatic cells of the *Drosophila* ovary. We find that depletion of some NPC subunits compromises the assembly of Yb-bodies and that loss of Nup54 and Nup58 specifically impacts *flam* export, but not that of bulk mRNAs. We show that Nup54 and Nup58 physically associate with Nxf1/Nxt1 as well as Yb, implying the existence of an export-coupled localisation mechanism specifying *flam* as a piRNA precursor. Considered together, our results suggest that genomic loci under strong selective pressure can co-opt NPC subunits to facilitate expression, thus expanding the repertoire of processes in which Nups play a role.

## Results

### FG nucleoporins Nup54 and Nup58 function specifically in transposon control

Various Nups have been ascribed gene regulatory functions, often via chromatin binding (*Strambio-De-Castillia et al., 2010*), and a subset of NPC subunits has been genetically implicated in transposon control in *Drosophila* ovaries (*Czech et al., 2013*; *Handler et al., 2013*; *Muerdter et al., 2013*; *Figure 1A*). To understand whether any of these Nups play a specific role in piRNA-guided TE silencing, we systematically assessed the effect of their depletion on cell viability, TE expression, and Yb-body formation in OSCs (*Figure 1A, B*). Knockdown of most Nups resulted in pronounced cell death and disassembled Yb-bodies, with little to no effect on TEs (*Figure 1B*, *Figure 1—figure supplement 2A–F*, *Figure 1—figure supplement 3*). Instead, loss of subunits of the Nup62 subcomplex (Nup54-Nup58-Nup62) and their scaffold Nup93-1 (*Chug et al., 2015*; *Stuwe et al., 2015*; *Ulrich et al., 2014*) caused strong TE de-repression (*Figure 1B*, *Figure 1—figure supplement 2A–F*). Among these, only the depletion of Nup54 and Nup58 resulted in TE up-regulation without severely affecting cell viability (*Figure 1B, Figure 1—figure supplement 2A*), potentially hinting to

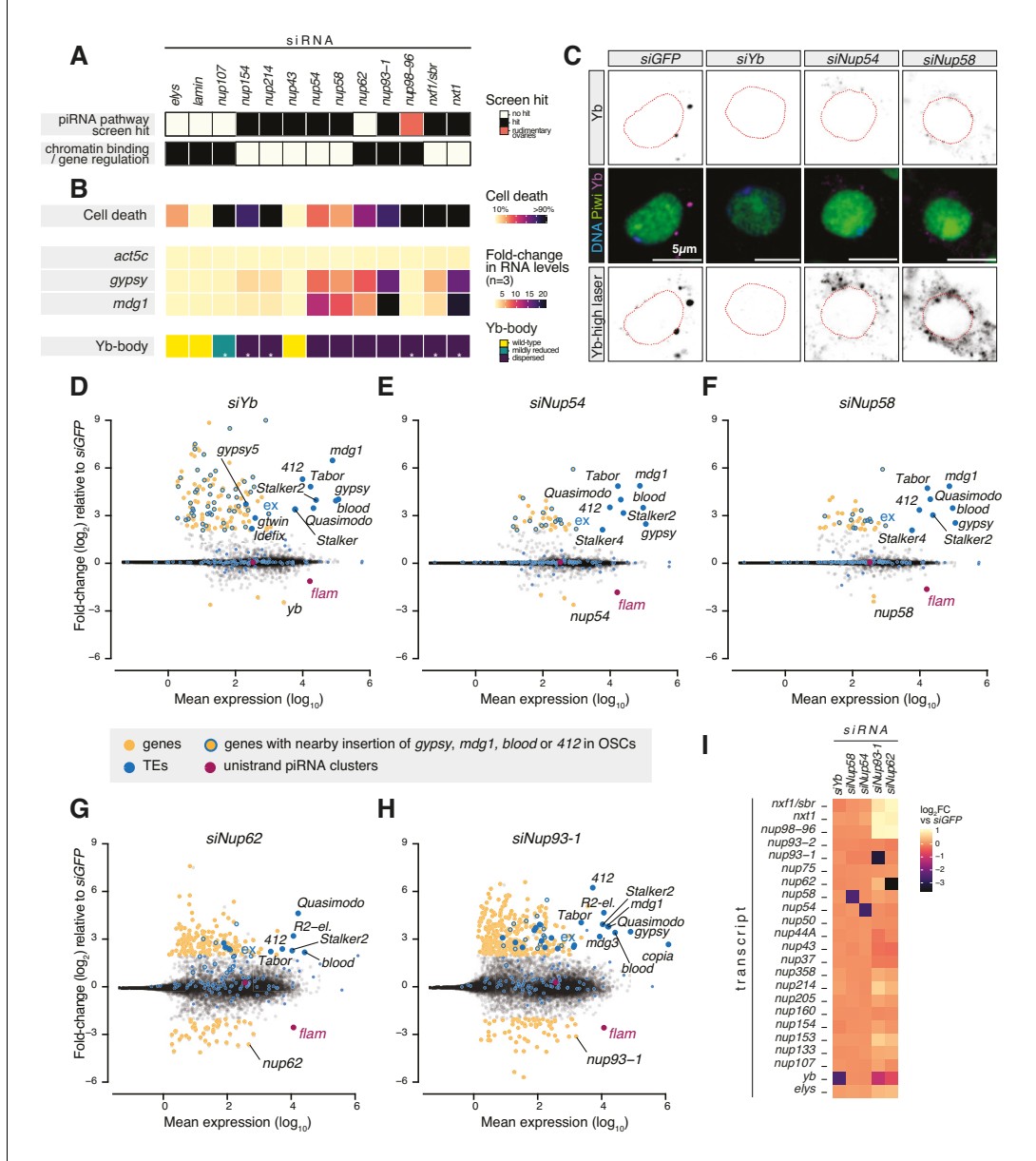

**Figure 1.** Requirement of Nuclear Pore Complex (NPC) subunits for gene regulation and transposon control. (A) Heatmap summarising literature data for selected Nups and export factors and (B) effects of each knockdown (kd) on cell viability, transposon expression levels with respect to *siGFP* and Yb-body assembly after 96 hr of siRNA treatment (n = 3; see also *Figure 1—figure supplement 2A–E* and *Figure 1—figure supplement 3*). Asterisks denote samples imaged after 48 hr of siRNA treatment because of lethality at later timepoints. (C) Confocal images of Yb and Piwi proteins in ovarian somatic cells (OSCs) upon the indicated kd. Dotted red lines denote nuclear envelope (see also *Figure 1—figure supplement 2F*). (D–H) MA plots showing mean expression levels (reads per million mapped reads, rpm) against fold-changes of genes and transposable elements (TEs) in RNA-seq from the indicated kd with respect to *siGFP* control (n = 4). Yellow: genes de-regulated more than fourfold with adjusted p value < 0.05; blue: TEs; blue outlines: genes de-regulated more than fourfold carrying a nearby TE (*gypsy, mdg1, 412,* or *blood*) insertion in OSCs; magenta: uni-strand piRNA clusters (*20A* and *flam*). (I) Fold-changes in transcript levels upon the indicated kd with respect to *siGFP*.

The online version of this article includes the following figure supplement(s) for figure 1:

**Figure supplement 1.** Yb-body formation is dependent on *flam*.

**Figure supplement 2.** Effects of Nuclear Pore Complex subunits on transposable element (TE) silencing and Yb-body assembly in ovarian somatic cells (OSCs).

**Figure supplement 3.** Effects of Nuclear Pore Complex subunits on Yb-body assembly in ovarian somatic cells (OSCs).

an effect distinct from general nuclear transport. Yb-bodies were also dispersed in *siNup54* and *siNup58* and residual Yb was visible only at increased laser power, despite an overall minor effect on Yb protein levels (*Figure 1B, C*, *Figure 1—figure supplement 2C–G*). Of note, TE de-repression caused by knockdown of *nup54* and *nup58* was comparable to that observed upon depletion of Nxt1 (*Figure 1B, Figure 1—figure supplement 2A*), reported to also function in the co-transcriptional gene silencing branch of the piRNA pathway (*Batki et al., 2019*; *Fabry et al., 2019*; *Murano et al., 2019*; *Zhao et al., 2019*). These data indicate that loss of TE silencing upon depletion of most Nups is likely a result of the general NPC function in gene expression, whereas Nup54 and Nup58 seem to have more specific roles in transposon control.

To test this hypothesis, we carried out RNA-seq from OSCs depleted of the Nup62 complex subunits (Nup62-Nup54-Nup58), the scaffold protein Nup93-1 or the piRNA biogenesis factor Yb (*Olivieri et al., 2010*; *Saito et al., 2010*; *Szakmary et al., 2009*). Cells depleted of Nup54 or Nup58 showed a strong increase in the expression of *gypsy*-family TEs, which are known to be expressed in the somatic compartment of the ovary and regulated by *flam* (*Figure 1D–F, Figure 1—figure supplement 1C–D*; *Lécher et al., 1997*; *Pélisson et al., 1994*; *Prud'homme et al., 1995*). Both the spectrum of TEs affected and the magnitude of de-repression were very similar to those observed in *yb* knockdowns. In contrast, we observed only a moderate impact on protein-coding genes, with 130 genes de-regulated in *siYb* (more than fourfold and adjusted p value < 0.05), 42 in *siNup54*, and 42 in *siNup58* (*Figure 1D–F, Figure 2—figure supplement 1A*). A substantial fraction of those genes up-regulated by more than fourfold is found nearby transposon insertions that become de-silenced when the piRNA pathway is compromised (49/126 or 39% in *siYb*, 16/39 or 41% in *siNup54*, 13/40 or 33% in *siNup58*) (*Figure 1D–F*). One such example is the *expanded* (*ex*) gene on chromosome 2L (*Figure 1D–F*). This strongly suggests that most of the gene expression changes observed upon these knockdowns are in fact a consequence of TE re-activation.

Although knockdown of *nup62* and *nup93-1* also caused de-repression of some *flam*-regulated TEs, we found much more pronounced mis-expression of protein-coding genes, with 207 genes de-regulated in *siNup62* and 417 in *siNup93-1* (more than fourfold and adjusted p value < 0.05) (*Figure 1G, H*), which could not be explained by proximity to nearby TE insertions. Furthermore, this was accompanied by de-repression of TEs that are not normally subject to piRNA-mediated silencing in somatic cells, for example, *R2-element* (*Figure 1G, H*). Of note, *siNup62* and *siNup93-1* resulted in expression changes of other Nups, RNA export factors, and Yb, presumably via indirect effects on nuclear transport and/or gene expression (*Figure 1I*). Considered together, our results indicate that, in somatic follicle cells, Nup54 and Nup58 play specialised roles dedicated to transposon silencing, distinct from Nup62 and Nup93-1. This functional specialisation of the two proteins, especially from their closest binding partners in the NPC, is highly surprising, particularly considering that Nup54 and Nup58 are integral components of an essential and ubiquitous protein complex and so presumed to have general functions across the animal.

## Nup54 and Nup58 disruption specifically affects *flam* transcript stability

Nup54 and Nup58 belong to the highly conserved class of 'FG-Nups' (*Beck and Hurt, 2017*; *Figure 2A*) and constitute the heterotrimeric Nup62 complex (Nup54-Nup58-Nup62) (*Chug et al., 2015*; *Stuwe et al., 2015*; *Ulrich et al., 2014*) that lines the inner channel of the NPC (*Beck and Hurt, 2017*; *Grandi et al., 1995*; *Kim et al., 2018*). The phenylalanine-glycine (FG) repeats of the Nup62 complex subunits contribute to the selective permeability barrier of the NPC and interact with nuclear transport receptors, such as Nxf1 (*Köhler and Hurt, 2007*). Our RNA-seq showed a reduction in steady-state levels of *flam* transcript upon *nup54*, *nup58*, *nup62*, *nup93-1*, and *yb* knockdowns in OSCs (*Figure 1D-H*, *Figure 2B*, *Figure 2—figure supplement 1B*). This was both specific to *flam* as other somatic piRNA source loci (e.g. the protein-coding gene *tj* or the piRNA cluster *20A*) were unaffected (*Figure 2—figure supplement 1C, D*), and unexpected since prior studies had shown accumulation of *flam* transcripts in cases where its conversion into piRNAs was disrupted, for example, by knockdown of the ribonuclease *zuc* (*Murota et al., 2014*). To probe the underlying mechanism, we first asked whether the impact was uniform throughout the locus. To address this question, we divided the *flam* genomic region into non-overlapping 1 kb bins and extracted those reads that could be mapped with high confidence (see Materials and methods). Plotting fold-changes in these 1 kb bins following *zuc* knockdown, which prevents *flam* processing into piRNAs, showed an increased precursor abundance that was uniform across the entire locus

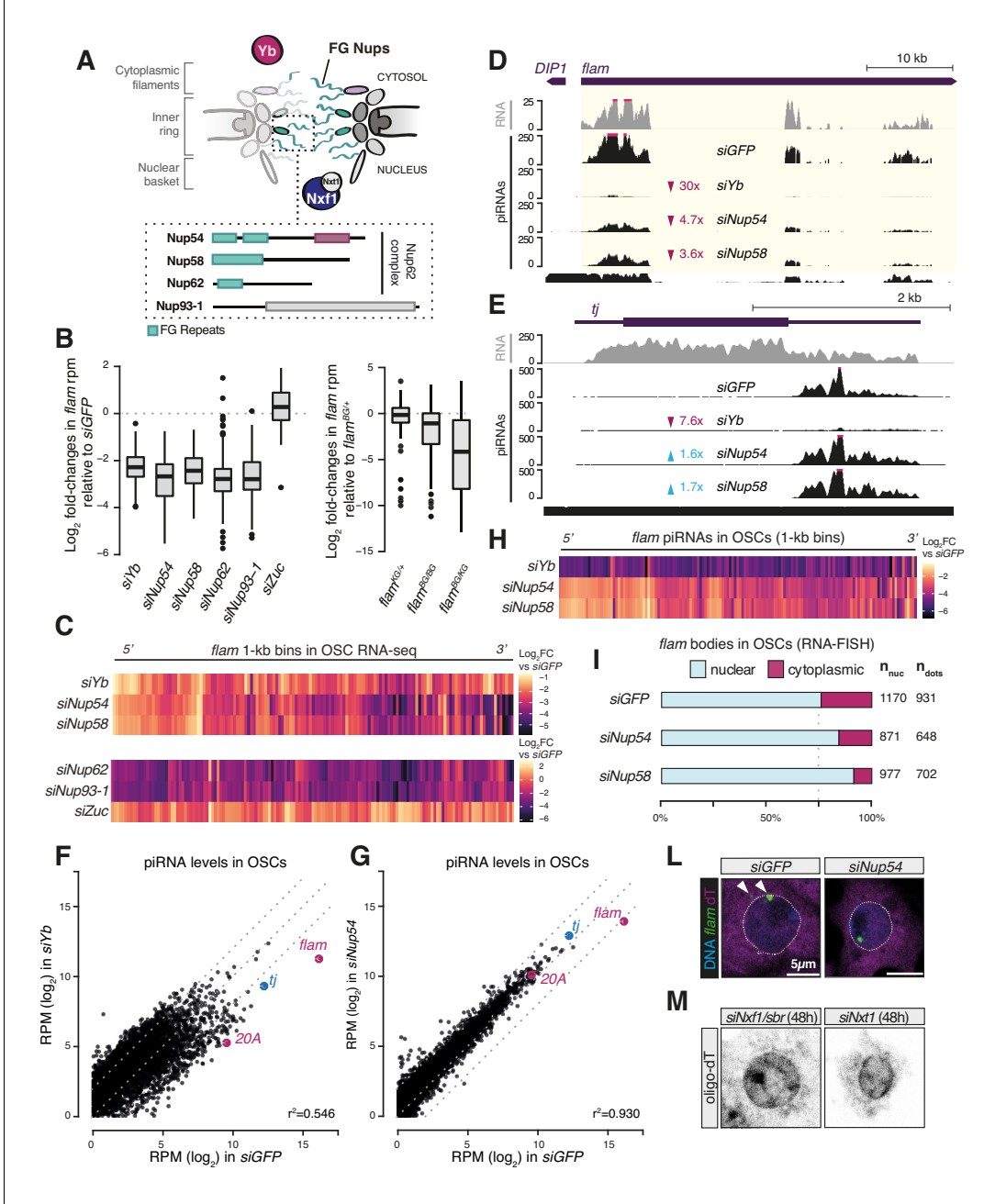

**Figure 2.** Phenylalanine-glycine (FG) nucleoporins Nup54 and Nup58 function specifically in transposable element (TE) silencing. (**A**) Cartoon showing the Nuclear Pore Complex (NPC) structure with nuclear and cytosolic factors involved in *flam* export; inset shows the domain structure of the Nup62 complex subunits and of Nup93-1. Green boxes: FG-repeats; purple box: Nup54-family domain; grey box: Nic96-family domain. (**B**) Box plots showing changes in *flam* RNA levels in the indicated knockdowns (kd) or genotypes. Each datapoint corresponds to a 1 kb bin. Fold-changes were calculated for each bin with respect to the *siGFP* control. (**C**) Heatmaps showing changes in *flam* RNA levels in the indicated kd. Each datapoint corresponds to a 1 kb bin, ordered from 5′ to 3′. Fold-changes were calculated for each bin with respect to the relative *siGFP* control. (**D, E**) Coverage plots of piRNAs and RNA-seq reads uniquely mapped to the *flam* locus or to the protein-coding gene *tj* upon the indicated kd. The mappability for 25 bp reads is shown at the bottom. (**F, G**) Scatter plots showing expression levels of PIWI-interacting RNAs (piRNAs) upon the indicated kd. (**H**) Heatmaps showing changes in *flam*-derived piRNA levels upon the indicated kd. Each datapoint corresponds to a 1 kb bin, ordered from 5′ to 3′. Fold-changes were calculated for each bin with respect to the *siGFP* control. (**I**) Quantification of nuclear and cytoplasmic *flam* RNA-FISH foci is shown. $N_{nuc}$ = total number of nuclei analysed; $n_{dots}$ = total number of *flam* foci counted. (**L**) Confocal images of *flam* RNA and polyA-tail containing transcripts (oligo-dT) in ovarian somatic cells (OSCs) upon the indicated kd (full panel in *Figure 2—figure supplement 3A*). Arrowheads indicate cytosolic *flam* foci. Dotted line denotes the nuclear envelope based on anti-lamin staining. (**M**) Confocal images of polyA-tailed transcripts (oligo-dT) in OSCs upon the indicated kd (full panel in *Figure 1—figure supplement 2C* and *Figure 1—figure supplement 3*).

*Figure 2 continued on next page*

*Figure 2 continued*

The online version of this article includes the following figure supplement(s) for figure 2:

**Figure supplement 1.** Effects of Nup54 and Nup58 knockdown (kd) on *flam* expression.

**Figure supplement 2.** Nup54 and Nup58 are required for PIWI-interacting RNA (piRNA) production from *flam*.

**Figure supplement 3.** Nup54 and Nup58 are required for *flam* export.

(*Figure 2B, C*). In contrast, depletion of Yb, Nup54, and Nup58 revealed a reduction that was more pronounced towards the 3′ end of *flam* (*Figure 2B, C*). In *nup62* and *nup93-1* knockdowns, reduced RNA levels were uniform, again highlighting a different role for these Nups (*Figure 2B, C*). As expected, we observed uniformly reduced RNA levels in *flam*^BG/KG^ trans-heterozygous mutants that impair transcription of the entire locus (*Figure 2B, Figure 2—figure supplement 1E*; *Brennecke et al., 2007*; *Goriaux et al., 2014*; *Malone et al., 2009*). All these analyses showed similar patterns using 100 kb sliding windows (*Figure 2—figure supplement 1F*), thus our results were consistent regardless of the window size.

Next, to determine whether the observed reduction stems from decreased transcription initiation, we examined nascent RNA at the *flam* locus via PRO-seq. PRO-seq in control cells (*siGFP*) revealed one major transcription initiation peak, as expected (*Brennecke et al., 2007*; *Goriaux et al., 2014*), and detected a second, previously unidentified, minor peak ~1 kb further downstream (*Figure 2—figure supplement 1G*). Knockdown of *nup54*, *nup58,* or *yb* had little to no effect on either signal around the transcription initiation site or within the first 10 kb and 40 kb of *flam*, whereas a more pronounced decrease in *siYb* cells was observed upon inspection of the entire locus (*Figure 2—figure supplement 1G, H*). In contrast, global PRO-seq signal from protein-coding genes and cluster *20A* was unchanged (*Figure 2—figure supplement 1I*). Overall, these observations are consistent with a hypothesis that loss of Nup54 or Nup58 reduces the stability of *flam* transcripts, with larger effects on regions distal from the transcription initiation sites.

## Nup54 and Nup58 are specifically required for *flam* export and processing into piRNAs

To analyse the effects of Nup54 and Nup58 depletion on piRNA populations, we sequenced small RNAs from OSC knockdowns. As previously reported (*Hirakata et al., 2019*), depletion of Yb caused a collapse in the antisense, TE-targeting piRNA population, but leaving 21-nt siRNAs unaffected (*Figure 2—figure supplement 2A, B*). Knockdown of *nup54* and *nup58* resulted in a approximately threefold decrease in antisense piRNAs, but this impact was highly specific to those derived from *flam* (*Figure 2D*, *Figure 2—figure supplement 2A, B*). In contrast, piRNAs derived from *tj* and cluster *20A* were unaffected or slightly more abundant (*Figure 2E*, *Figure 2—figure supplement 2C*). Whilst *siYb* had a general impact on piRNA production, in line with its role as key biogenesis factor (*Hirakata et al., 2019*; *Ishizu et al., 2015*; *Ishizu et al., 2019*; *Murota et al., 2014*; *Olivieri et al., 2010*; *Saito et al., 2010*), *siNup54* and *siNup58* only showed a reduction of *flam*-derived piRNAs (*Figure 2F, G*, *Figure 2—figure supplement 2D*). Binning analysis of the *flam* locus showed that piRNAs were lost homogenously along the entire cluster in *siYb* (*Figure 2H, Figure 2—figure supplement 2E*), unlike the precursor transcript levels measured by RNA-seq, indicating that no processing can occur in the absence of Yb. In contrast, piRNA loss upon *nup54* and *nup58* knockdown was more pronounced towards the 3′ region (*Figure 2H, Figure 2—figure supplement 2E*), mirroring the precursor transcript reduction observed by RNA-seq. These data are in agreement with a defect in precursor specification upon *siYb* and suggest that Nup54 and Nup58 play a role in *flam* piRNA biogenesis that is distinct from that of Yb.

RNA-FISH for *flam* in OSCs typically shows discrete foci on the nuclear rim and in the cytosol (*Dennis et al., 2016*; *Murota et al., 2014*; *Figure 2L, Figure 2—figure supplement 3A*). Depletion of Nup54 and Nup58 resulted in clustering of the signal in one predominant focus within the nuclear envelope (*Figure 2I–L, Figure 2—figure supplement 3A*). Since the *flam* DNA locus is located at the nuclear periphery (*Figure 2—figure supplement 3B, C*), this RNA nuclear focus likely corresponds to *flam* RNPs stalled prior to nuclear export (*Dennis et al., 2016*; *Dennis et al., 2013*). Nonetheless, this positions the *flam* locus in close proximity to the NPC, possibly underscoring the need of a specialised transcription-coupled export machinery linked directly to piRNA production.

Of note, neither *nup54* nor *nup58* knockdown affected the distribution of bulk polyadenylated mRNAs (*Figure 1—figure supplement 2C*, *Figure 1—figure supplement 3*, *Figure 2L*, *Figure 2—figure supplement 3A*), unlike depletion of Nxf1 or Nxt1, which instead resulted in nuclear retention of newly synthesised mRNA (*Figure 1—figure supplement 2C*, *Figure 1—figure supplement 3*, *Figure 2M*), as reported (*Herold et al., 2001*).

Considered together, these results indicate that transposon silencing defects resulting from loss of Nup54 or Nup58 arise from their role in facilitating *flam* export from the nucleus. In OSCs and follicle cells of the ovary, this activity dominates any general function in NPC biology since cells are viable and distributions of bulk mRNAs are largely unaffected upon their depletion. Given that the effect is most prominent on the 3′ end of the transcript, we hypothesise that Nup54 and Nup58 might be required to ensure processivity of nuclear export of this, otherwise unstable, long transcript. In this scenario, residual *flam* molecules (likely corresponding to the 5′ portion of the cluster) that reach the cytosol upon *siNup54* and *siNup58* might still be processed by Yb, although with lower efficiency than within properly formed Yb-bodies. This role of Nup54 and Nup58 could also, directly or indirectly, affect transcriptional elongation and termination of the piRNA cluster; however, further work will be required to test this hypothesis.

## Characterisation of Nup54 and Nup58 domains required for TE silencing

The FG-repeats of Nup54 and Nup58 protrude into the inner channel to form the NPC permeability barrier and interact with nuclear transport receptors, such as Nxf1, making these regions obvious candidates for regulating *flam* export. We therefore designed deletion mutants targeting Nup54 and Nup58 domains and assayed their ability to interact with other Nups and to rescue TE de-repression in OSCs (*Figure 3A*). These constructs lack either the amino-terminal region, which in both Nups carries the FG-repeats, or the carboxy-terminal part, which mediates the interaction of these proteins with each other and with the rest of the pore (*Figure 3A, B*; *Chug et al., 2015*; *Stuwe et al., 2015*). We depleted Nup54 or Nup58 individually in OSCs and then re-introduced either an siRNA-resistant full-length (FL) or deletion construct of Nup54/Nup58, or a negative control (mCherry), and assayed their ability to restore transposon repression. As expected, FL Nup54 and Nup58 rescued *mdg1* up-regulation to levels comparable to *siGFP* (*Figure 3C*). Likewise, deleting the FG-repeats in Nup54 and Nup58 (ΔFG) had little effect on their TE silencing capability compared to FL Nups. In contrast, Nup54 and Nup58 lacking the C-terminal domain (ΔC) failed to rescue transposon de-repression (*Figure 3C*, *Figure 3—figure supplement 1A*). These results suggest that the ability to interact with the other Nups is required to ensure TE silencing, and thus that Nup54 and Nup58 carry out this function from within the NPC.

With very few exceptions, Nup null mutants are generally not viable in *Drosophila*. One such exception, the *nup54^{MB003363}* (*nup54^{MB}*) allele, produces a truncated protein lacking the carboxy-terminal region due to a *Mi{ET1}* transposon insertion (*Nallasivan et al., 2020*). This shortened Nup54 protein lacks the Nup54-family domain (*Figure 3A, D, Figure 3—figure supplement 1B*) and fails to co-precipitate with Nup58 (*Figure 3E*), as expected from earlier reports (*Chug et al., 2015*). Thus, it resembles the carboxy-terminal truncation (Nup54^{ΔC}) that is unable to sustain transposon repression in OSCs (*Figure 3A–C*). We sought to determine if this hypomorphic allele phenocopies the molecular phenotype of the *nup54* knockdown in OSCs. Trans-heterozygous flies (*nup54^{MB/9B4}*) carrying the *nup54^{MB}* allele over the *nup54^{9B4}* deficiency spanning the entire *nup54* locus (*Nallasivan et al., 2020*) were viable, although at reduced Mendelian ratios, and had smaller, but not rudimentary, ovaries (*Figure 3—figure supplement 1C*). To minimise differences in TE content between fly strains, we crossed the *nup54^{MB}* allele to *w^{1118}* flies and compared *nup54^{MB/9B4}* trans-heterozygote mutants to *nup54^{MB/w1118}* heterozygotes and to *w^{1118}* controls. RNA-seq from ovaries of trans-heterozygous *nup54^{MB/9B4}* flies showed TE de-repression and reduced *flam* RNA levels (*Figure 3F, G*, *Figure 3—figure supplement 1D*). Expression levels of major dual-strand piRNA clusters (*42AB*, *38C*, and *80F*), which rely on different, specialised transcription and export pathways, were unchanged, if not slightly higher, in the case of *80F* (*Figure 3F*). Of note, the increased expression of *flam*-regulated transposons (e.g. *gypsy*) was already evident in *nup^{MB/w1118}* heterozygous flies (*Figure 3G*, *Figure 3—figure supplement 1D*). In this in vivo setting, we observed a broader impact on the expression of protein-coding genes than for knockdowns in cell culture (*Figure 3—figure supplement 1D, E*), likely reflecting a more general function of Nup54 in the germline tissue of the

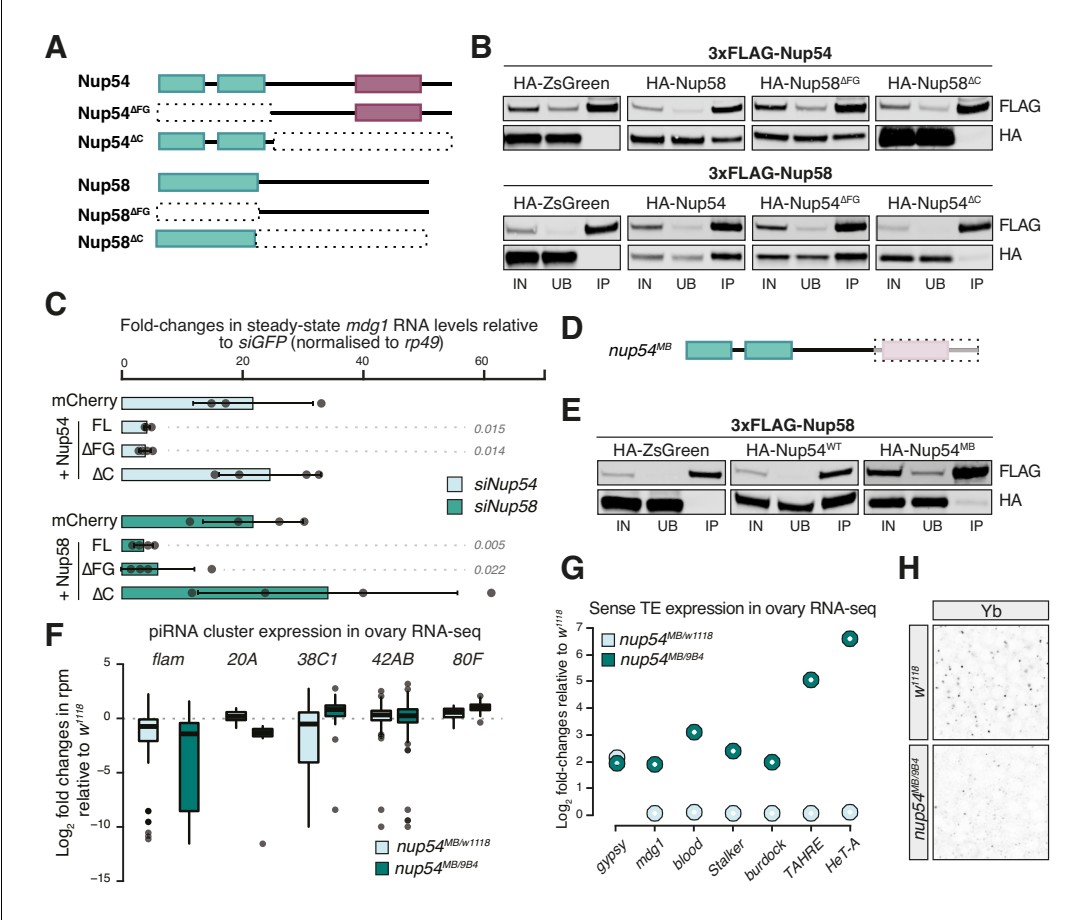

**Figure 3.** Nup54 and Nup58 are specifically required for *flam* export and processing into PIWI-interacting RNAs (piRNAs). (A) Schematic of the Nup54 and Nup58 domain structure and the deletion constructs used in rescue experiments and co-immunoprecipitation assays. Green box: phenylalanine-glycine (FG)-repeats; purple box: Nup54-family domain. (B) Western blots of FLAG-tag co-immunoprecipitates from lysates of S2 cells transfected with the indicated constructs. IN: input; UB: unbound; IP: immunoprecipitate. (C) Fold-changes in steady-state RNA levels of the *mdg1* transposon in ovarian somatic cells (OSCs) nucleofected with the indicated siRNAs and rescue constructs. Values are relative to *siGFP* and normalised to *rp49*. Error bars indicate standard deviation (n = 4). p values indicated next to each bar were calculated with respect to the relative mCherry control (unpaired *t*-test). (D) Schematic of the *nup54^{MB}* allele. The dashed box indicates the portion that is absent in the mutant (purple box: Nup54 family domain). (E) Western blots of FLAG-tag co-immunoprecipitates from lysates of S2 cells transfected with the indicated constructs. (F) Box plots showing changes in piRNA cluster transcript levels in the indicated genotypes. Each datapoint corresponds to a 1 kb bin. (G) Steady-state levels of transposable element (TE) transcripts in the indicated genotypes. (H) Confocal images of Yb protein in follicle cells of the indicated genotypes (full panel in *Figure 3—figure supplement 2*).

The online version of this article includes the following figure supplement(s) for figure 3:

**Figure supplement 1.** Effect of *nup54^{MB}* mutants on gene expression.

**Figure supplement 2.** Effect of *nup54^{MB}* mutants on Yb-body assembly.

ovary. In this regard, also germline TEs were up-regulated in *nup54* mutants (*Figure 3G, Figure 3—figure supplement 1D*), confirming an earlier genetic link to TE silencing in germ cells (*Czech et al., 2013*). Close inspection of the levels of *flam* RNAs using the previously described binning strategy showed that the down-regulation was more prominent towards promoter-distal regions of the cluster (*Figure 3—figure supplement 1F*), thus recapitulating our results from knockdowns in OSCs. Lastly, Yb-bodies were reduced in *nup54^{MB/9B4}* trans-heterozygous flies (*Figure 3H, Figure 3—figure supplement 2*). The *yb* transcript is mildly up-regulated in *nup54^{MB/9B4}* (*Figure 3—figure supplement 1E*), thus loss of Yb-bodies likely arises as a result of compromised *flam* export.

Overall, these data confirm a requirement of Nup54 and Nup58 for the expression and TE silencing activity of *flam* in both OSCs and in vivo. We find that the ability of Nup54 and Nup58 to form a complex is critical for their function but the integrity of the FG-repeat regions of each protein

individually is not. Of note, the FG-repeats of their yeast homologs Nup49 and Nup57 are also dispensable for cell survival, leading to the suggestion that not all FG-Nups are equivalent and some can facilitate distinct translocation pathways (*Iovine et al., 1995*; *Strawn et al., 2004*).

## Nup54/Nup58 coordinate nuclear export and cytosolic licensing of *flam* via Nxf1 and Yb

To understand the molecular basis of this specificity, we explored three avenues: (1) physical proximity of the *flam* genomic locus to cytosolic Yb-bodies, (2) direct binding of Nup54 and/or Nup58 to *flam* RNA, or (3) an interaction between Nup54/Nup58 and *flam* RNP export complexes. Although the *flam* locus is at the nuclear periphery, we failed to detect any consistent correlation with the position of Yb-bodies on the opposite side of the nuclear envelope in both OSCs and ovaries (*Figure 2—figure supplement 3C*; *Figure 2—figure supplement 3D* ) thus the first hypothesis seems unlikely. To test the second possibility, we carried out CLIP-seq for Nup54, Nup58, and the mRNA exportin Nxf1, which was previously implicated in *flam* export (*Dennis et al., 2016*). Nxf1 was shown to be loaded onto nascent mRNAs upon splicing, to interact with FG-Nups and to function itself as a mobile Nup (*Ben-Yishay et al., 2019*; *Derrer et al., 2019*; *Köhler and Hurt, 2007*), therefore representing the most probable link between the nascent *flam* transcript and the NPC. CLIP-seq experiments with HALO-tagged Nup54 and Nup58 failed to enrich for *flam* RNA, whereas independent experiments with amino- and carboxy-terminally HALO-tagged Nxf1 showed an interaction towards the 5′ end of the *flam* transcript (*Figure 4—figure supplement 1A*), which was reported to undergo splicing (*Goriaux et al., 2014*). Since the association of Nxf1 with a cargo transcript is believed to follow a splicing event, this may implicate its co-transcriptional loading onto the 5′ spliced region of *flam* as the initial signal for export. So far, no splicing has been reported in the downstream regions of *flam*.

Next, we sought to determine whether the so far identified factors participating in *flam* export (i.e. Nxf1, Nup54, Nup58, and Yb) physically interact with each other. We first searched for protein interactions between Yb and the NPC via BASU-mediated proximity labelling, followed by mass spectrometry (PL-MS) in OSCs (*Kim et al., 2014*; *Munafò et al., 2019*; *Ramanathan et al., 2018*; *Roux et al., 2012*). In this experiment, the protein of interest is fused to a biotin ligase (BASU) and expressed in OSCs. Upon biotin supplementation, the fusion protein produces activated biotin-AMP intermediates that covalently attach to accessible lysine residues of proteins in close spatial proximity (*Roux et al., 2012*). Biotinylated proteins are subsequently recovered by streptavidin pulldown and identified by quantitative mass spectrometry. Yb PL-MS enriched, among others, for the known Yb interactors Armi and Piwi (*Hirakata et al., 2019*; *Saito et al., 2010*) but not for mitochondrial piRNA pathway proteins (*Figure 4—figure supplement 1B*, *Figure 4—figure supplement 1— source data 1*), in line with the current model for piRNA biogenesis that postulates shuttling of Armi between Yb-bodies and mitochondria but not that of Yb itself (*Ge et al., 2019*; *Munafò et al., 2019*; *Yamashiro et al., 2020*). Only one Nup, Nup214, which localises to the cytoplasmic filaments and whose yeast homolog (Nup159) cooperates with the DEAD-box helicase Dbp5 in disassembling export complexes, was detected as significantly enriched. This result indicates that Yb is not an integral NPC component but may nonetheless localise proximally to the cytosolic filaments of the pore. Instead, PL-MS for Nup54 and Nup58 (*Figure 4A, B*, *Figure 4—source data 1*, *Figure 4—source data 2*) enriched for various subunits of the NPC, including all the components of the Nup62 subcomplex and the scaffold Nup93-1. We detected a modest enrichment of Nxf1, which is known to interact with FG-Nups and to export *flam* RNA (*Dennis et al., 2016*; *Segref et al., 1997*). Notably, Yb was among the most highly enriched proteins in PL-MS for both Nups (*Figure 4A, B*), thus indicating that Yb contacts the (cytoplasmic side of the) NPC in OSCs. No other known piRNA pathway factor was detected in proximity to Nup54 and Nup58 with this approach.

Since neither of the Nups was detected by Yb PL-MS, we sought to probe a putative interaction between Yb and the NPC via alternative methods. Using immunofluorescence, we could often observe overlapping and/or adjacent signals between Yb and Nup54/Nup58 (*Figure 4C*), as approximated by biotinylation staining from TurboID fusion proteins (*Branon et al., 2018*); however, the Nup signal was found along the entire nuclear envelope and not exclusively in association with Yb-bodies (*Figure 4C*). Conversely, Yb-bodies were often distinct from Nup foci, thus indicating that Yb is not stably anchored to the NPC but rather dynamic and possibly explaining why only the cytosolic Nup214 was detected by Yb PL-MS. These data were not an artefact due to protein over-expression

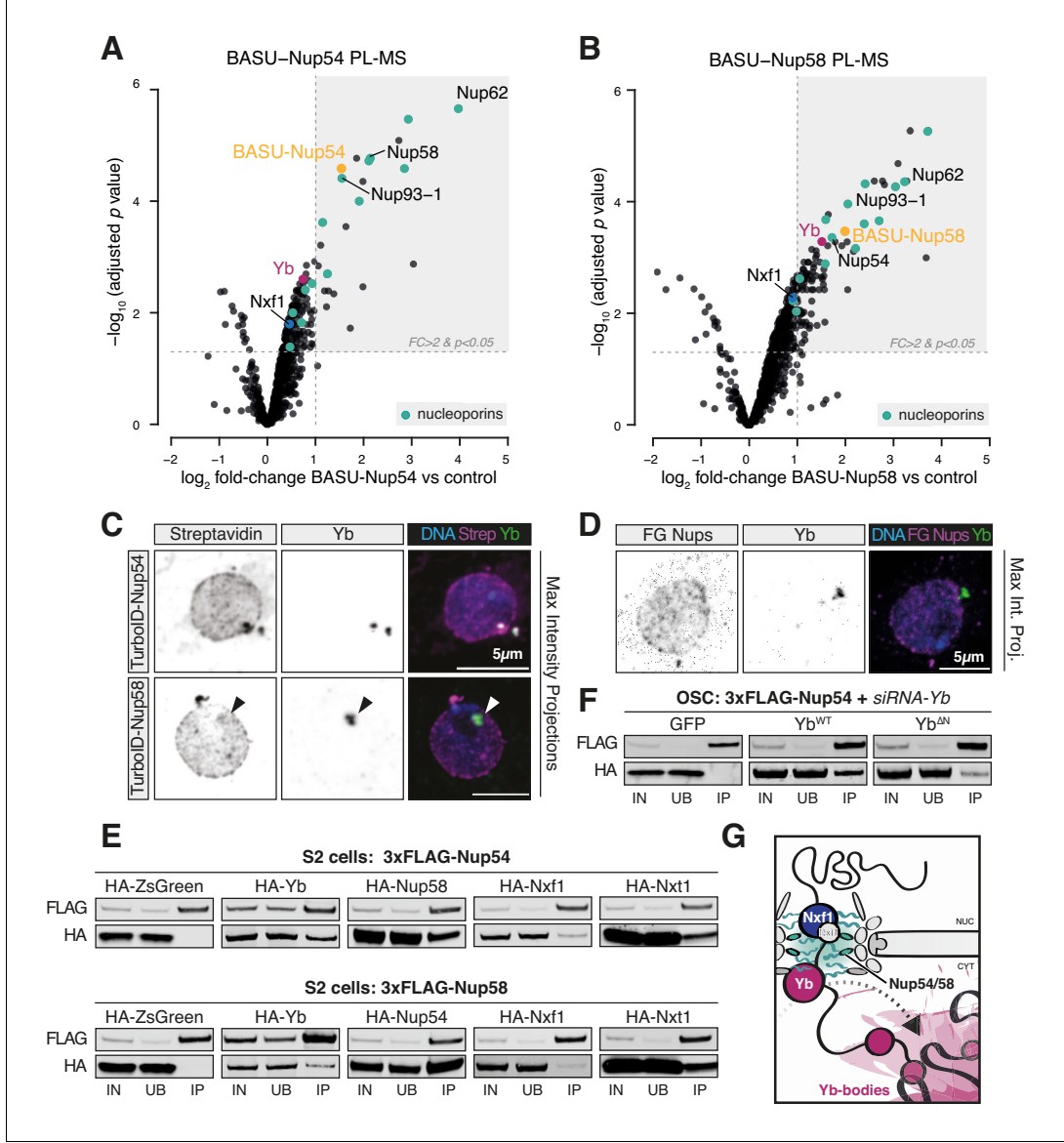

**Figure 4.** Nup54 and Nup58 coordinate export and licensing of *flam* via Nxf1 and Yb. (**A, B**) Volcano plots showing enrichment and corresponding significance of biotinylated proteins identified via proximity labelling, followed by mass spectrometry (PL-MS) from ovarian somatic cells (OSCs) expressing the indicated constructs against control (n = 3). NPC subunits: green; Yb: magenta; Nxf1: blue; bait protein: yellow. (**C, D**) Confocal images of Yb protein and TurboID-Nup54/Nup58 or phenylalanine-glycine nucleoporins (FG-Nups) in OSCs. Arrowheads indicate Yb-bodies juxtaposed to Nup foci. (**E, F**) Western blots of FLAG-tagged co-immunoprecipitates from cells transfected with the indicated constructs and siRNAs. (**G**) Proposed model of *flam* export-coupled licensing in OSCs.

The online version of this article includes the following source data and figure supplement(s) for figure 4:

**Source data 1.** Source data for volcano plot shown in *Figure 4A*.
**Source data 2.** Source data for volcano plot shown in *Figure 4B*.
**Figure supplement 1.** Nup54/Nup58 coordinate *flam* export and licensing by bridging Nxf1 and Yb.
**Figure supplement 1—source data 1.** Source data for volcano plot shown in *Figure 4—figure supplement 1B*.

as we observed similar staining patterns using an antibody specific for FG-Nups (*Figure 4D, Figure 4—figure supplement 1C*). This suggests that physical associations between Yb and Nup54/Nup58 within the NPCs are likely transient and that Nup54/Nup58 are not confined only to a discrete site on the nuclear envelope. With Nup54 and Nup58 expected to being present in every

pore, *flam* export might only be initiated where they contact Yb, and this in turn would nucleate Yb-body assembly. We cannot formally exclude the hypothesis that a pool of Nup54 and Nup58 is also present within Yb-bodies and carries out its function independently of the NPC, although in this case one might have expected them to be detected in Yb PL-MS experiments.

To test whether Yb binds directly to Nup54 and Nup58, we carried out co-immunoprecipitation experiments in *Drosophila* Schneider 2 (S2) cells, which do not express a functional piRNA pathway and represent a convenient system to investigate protein-protein interactions that are not bridged by other piRNA pathway components. We used 3xFLAG-tagged Yb, Nup54, or Nup58 as baits and probed for the corresponding HA-tagged versions. As expected, Nup54 and Nup58 recovered a substantial amount of their respective partner Nup (*Figure 4E*). We also detected interactions with Yb and, to a lesser extent, with Nxt1 and Nxf1 (*Figure 4E*). Co-immunoprecipitation of Nup54/Nup58 and Yb in OSCs was insensitive to RNase I treatment (*Figure 4—figure supplement 1D*), thus indicating that their interaction is not mediated by *flam* RNA, and to knockdown of endogenous Yb, thus ruling out over-expression artefacts (*Figure 4F, Figure 4—figure supplement 1F*). Reciprocal pull-down experiments in OSCs confirmed these findings, with 3xFLAG-Yb recovering HA-tagged Nup54 and Nup58 and only modest amounts of Nxt1, but not Nxf1 (*Figure 4—figure supplement 1E*). The amino-terminal region of Yb contains a HelC domain, which is required exclusively for *flam* piRNA production as opposed to piRNA biogenesis in general (*Hirakata et al., 2019*). Its deletion in OSCs recapitulates the biogenesis phenotype caused by *nup54* and *nup58* knockdown. We observed that this deletion weakened, though did not completely abolish, the interaction between Yb and the Nups (*Figure 4F, Figure 4—figure supplement 1F*), thus indicating that it is at least partially responsible for anchoring Yb to the NPC. Co-immunoprecipitation of HA-tagged FL Nup54/Nup58 or versions carrying the FG or carboxy-terminus domain deletions showed that Yb interacts with the FG-Nups only when they are able to assemble into the NPC as deletion of either Nup carboxy-terminus, which ablates interaction between the two Nups (*Figure 3B*), reduced the interaction (*Figure 4—figure supplement 1G*).

Although we cannot exclude the involvement of additional adaptor proteins that remained undetected by our approach, these data suggest that Yb associates with the cytoplasmic side of the NPC and binds to exiting *flam* transcripts. Because binding to RNA triggers the aggregation of *flam*-Yb RNPs into phase-separated Yb-bodies (*Hirakata et al., 2019*), we hypothesise that this provides the directionality to *flam* transport.

## Discussion

Here, we find that Nup54 and Nup58 are specifically required for TE regulation in the follicle cells of *Drosophila* ovaries by enabling export and subsequent piRNA production from *flam* transcripts. This functional requirement is distinct from the general role of the NPC as other Nups, even those most proximal to Nup54 and Nup58 within the pore, are broadly required for gene expression and cell survival. These findings consolidate our view of Nups as dynamic players in various cellular processes and expand the variety of roles ascribed to the Nups.

Though *flam* bears canonical features common to other mRNAs, it is nonetheless specifically recognised and processed into piRNAs. A dedicated export-coupled licensing, involving Nup54 and Nup58, may allow this long transcript to be escorted directly from its genomic origin to the Yb-bodies to facilitate piRNA production. If this machinery is disrupted, for example, by loss of Nup54 and Nup58, *flam* transcripts are confined to the nucleus and destabilised, thus underscoring the need for a unified process from transcription to licensing. It is interesting to note that *siYb* caused a slight decrease of PRO-seq signal across the entire piRNA cluster locus, possibly indicating that disruption of *flam* export-coupled licensing negatively affects its transcription via a yet-unknown feedback mechanism. We find that Nup54 and Nup58 interact with both the nuclear (Nxf1/Nxt1) and cytosolic (Yb) components of *flam* expression and thus suggest that these factors bridge nuclear export to cytosolic fate specification. Although several aspects of the molecular mechanism remain to be elucidated (i.e. how this links to upstream transcription and whether additional factors are involved in the nucleus or in the cytosol), we propose a tentative model whereby this Nxf1-NPC-Yb axis coordinates initiation, processivity, and directionality of *flam* export (*Figure 4G*), directly feeding the transcript into the piRNA biogenesis route via Yb-bodies. Our rescue experiments and protein-protein interaction studies argue for this TE silencing function of Nup54 and Nup58 to be carried out from within

the pore, especially since neither Nup is among the most dynamic components of the NPC (*Rabut et al., 2004*). However, we cannot completely exclude a function for Nup54 and Nup58 in the cytosol, possibly as components of Yb-bodies. We find that Nup54 and Nup58 interact directly with both Yb and Nxf1; however, it is presently unclear how the transcript is released from the Nxf1/Nxt1 export complexes and handed over to Yb, especially since we did not detect physical interactions between Yb and Nxf1. It cannot be excluded that such interaction exists and remained undetected in our over-expression experiments or that additional adaptor proteins take part in this process. Alternatively, Nup54/58 may form independent complexes with Yb and Nxf1; however, our lack of enrichment for *flam* transcripts associated with Nup54/58 makes this seem unlikely. Super-resolution imaging of the NPC will be required to clarify the relative position of each component of this export route and to understand whether Nup54 and Nup58 function also outside the NPC. Of note, cytosolic Nup358 has been shown to be required for piRNA production from the dual-strand cluster *42AB* in nurse cells (*Parikh et al., 2018*). Its depletion leads to prominent de-localisation of Piwi, which we did not observe upon loss of Nup54/Nup58, thus suggesting a different mode of action. Nonetheless, this further underscores that different NPC subunits can be co-opted for TE silencing in tissue-specific contexts.

The proposed mechanism for *flam* export-coupled licensing relies on a tissue-specific effector (Yb) and could represent a broader paradigm for co-option of FG-Nups for specific transcript trafficking routes. We envision that in principle any transcript subject to strong selective pressure could evolve a dedicated export machinery via adapting FG-Nup functions in a cell-type-specific manner. Such mechanisms may have gone unnoticed previously because of the general role of the NPC. Interestingly, several Nup genes in *Drosophila* show signs of rapid adaptive evolution that result in hybrid incompatibilities (*Presgraves et al., 2003*; *Tang and Presgraves, 2009*), which is often a hallmark of genes involved in genetic conflicts, such as transposon control. We further speculate that a tissue-specific control of export routes might contribute to explain the molecular mechanisms underlying so-called 'nucleoporopathies', human syndromes caused by mutated Nups (*Beck and Hurt, 2017*; *Braun et al., 2018*; *Miyake et al., 2015*). In these diseases, mutation or expression changes of a Nup present in all cells of the organism leads to tissue-specific phenotypes. We hypothesise that this might stem from specific roles of those Nups in regulating genes that are essential for the functionality of that particular tissue, which would in turn make the said tissue especially susceptible to the loss of the Nup. Future investigation will shed light on how widespread these mechanisms might be.

# Materials and methods

**Key resources table**

| Reagent type (species) or resource | Designation | Source or reference | Identifiers | Additional information |
|---|---|---|---|---|
| Gene (*Drosophila melanogaster*) | *fs(1)Yb* | FlyBase | FBgn0000928 | |
| Gene (*Drosophila melanogaster*) | *flamenco* | FlyBase | FBgn0267704 | |
| Gene (*Drosophila melanogaster*) | *Nup54* | FlyBase | FBgn0033737 | |
| Gene (*Drosophila melanogaster*) | *Nup58* | FlyBase | FBgn0038722 | |
| Gene (*Drosophila melanogaster*) | *Elys* | FlyBase | FBgn0031052 | |
| Gene (*Drosophila melanogaster*) | *Nup43* | FlyBase | FBgn0038609 | |
| Gene (*Drosophila melanogaster*) | *Nup214* | FlyBase | FBgn0010660 | |
| Gene (*Drosophila melanogaster*) | *Nup62* | FlyBase | FBgn0034118 | |

*Continued on next page*

*Continued*

| Reagent type (species) or resource | Designation | Source or reference | Identifiers | Additional information |
|---|---|---|---|---|
| Gene (*Drosophila melanogaster*) | *Nup93-1* | FlyBase | FBgn0027537 | |
| Gene (*Drosophila melanogaster*) | *Nup98-96* | FlyBase | FBgn0039120 | |
| Gene (*Drosophila melanogaster*) | *Nxf1/Sbr* | FlyBase | FBgn0003321 | |
| Gene (*Drosophila melanogaster*) | *Nxt1* | FlyBase | FBgn0028411 | |
| Gene (*Drosophila melanogaster*) | *lamin* | FlyBase | FBgn0002525 | |
| Gene (*Drosophila melanogaster*) | *Nup107* | FlyBase | FBgn0027868 | |
| Gene (*Drosophila melanogaster*) | *Nup154* | FlyBase | FBgn0021761 | |
| Gene (*Drosophila melanogaster*) | *Zuc* | FlyBase | FBgn0261266 | |
| Gene (*Drosophila melanogaster*) | *tj* | FlyBase | FBgn0000964 | |
| Antibody | Anti-Piwi (Rabbit polyclonal) | DOI:10.1016/j.cell.2007.01.043 | | IF(1:500) WB(1:5000) |
| Antibody | Anti-Yb ( Mouse monoclonal) | DOI:10.1101/gad.1989510 | | IF(1:500) WB(1:1000) |
| Antibody | Anti-Yb (Rabbit polyclonal) | DOI:10.1038/emboj.2011.308 | | WB(1:1000) |
| Antibody | Anti-tubulin (Rabbit polyclonal) | Abcam | Cat# ab18251, RRID:AB_2210057 | WB(1:5000) |
| Antibody | Anti-HA (Rabbit monoclonal) | Cell Signaling Technology | Cat# 3724, RRID:AB_1549585 | WB(1:1000) |
| Antibody | Anti-HA (Rabbit polyclonal) | Abcam | Cat# ab9110, RRID:AB_307019 | WB(1:1000) |
| Antibody | Anti-FLAG (Mouse monoclonal) | Sigma | Cat# F1804, RRID:AB_262044 | WB(1:1000) |
| Antibody | Anti-lamin (Mouse monoclonal) | Developmental Studies Hybridoma Bank | Cat# adl67.10, RRID:AB_528336 | IF(1:200) |
| Antibody | Anti-Nuclear Pore Complex Proteins (Mouse monoclonal) | Biolegend | Cat# 902907, RRID:AB_2565026 | IF(1:500) |
| Antibody | Anti-Mouse IgG Alexa Fluor-488 (Goat polyclonal) | Thermo Fisher Scientific | Cat# A-11029, RRID:AB_2534088 | IF(1:500) |
| Antibody | Anti-Rabbit IgG Alexa Fluor-647 (Goat polyclonal) | Thermo Fisher Scientific | Cat# A-21245, RRID:AB_2535813 | IF(1:500) |
| Antibody | GFP-Booster Atto-488 (Alpaca monoclonal) | Chromotek | Cat# gba488-100, RRID:AB_2631386 | IF(1:500) |
| Commercial assay or kit | IRDye 800CW Streptavidin | LI-COR | Cat# 926-32230 | WB(1:4000) |
| Commercial assay or kit | Streptavidin, Alexa Fluor 555 Conjugate | Thermo Fisher Scientific | Cat# S-21381 | IF(1:500) |
| Commercial assay or kit | Mouse monoclonal anti-FLAG M2 magnetic beads | Sigma-Aldrich | Cat# CatM8823, RRID:AB_2637089 | |
| Commercial assay or kit | Dynabeads MyOne Streptavidin C1 | Thermo Fisher Scientific | Cat# 65001 | |
| Commercial assay or kit | Pierce IP Lysis Buffer-100 ml | Thermo Fisher Scientific | Cat# 87787 | |

*Continued on next page*

*Continued*

| Reagent type (species) or resource | Designation | Source or reference | Identifiers | Additional information |
|---|---|---|---|---|
| Commercial assay or kit | RIPA Lysis and Extraction Buffer | Thermo Fisher Scientific | Cat# 89901 | |
| Commercial assay or kit | Thermo Scientific Pierce anti-HA magnetic beads | Thermo Fisher Scientific | Cat# 88836 | |
| Commercial assay or kit | cOmplete, Mini, EDTA-free Protease Inhibitor Cocktail | Sigma-Aldrich | Cat# 11836170001 | |
| Commercial assay or kit | RNasin Plus RNase Inhibitor | Promega | Cat# N2615 | |
| Commercial assay or kit | Effectene Transfection Reagent | Qiagen | Cat# 301427 | |
| Commercial assay or kit | Nucleofector Kit V | Lonza | Cat# VVCA-1003 | |
| Commercial assay or kit | DNase I, Amplification Grade | Thermo Fisher Scientific | Cat# 18068015 | |
| Commercial assay or kit | RNaseOUT Recombinant Ribonuclease Inhibitor | Thermo Fisher Scientific | Cat# 10777019 | |
| Commercial assay or kit | Magne HaloTag Beads, 20% Slurry | Promega | Cat# G7282 | |
| Commercial assay or kit | Deoxynucleotide Solution Set (100 mM; 25 µmol each) | New England Biolabs | Cat# N0446S | |
| Commercial assay or kit | SuperScript III Reverse Transcriptase | Thermo Fisher Scientific | Cat# 18080085 | |
| Commercial assay or kit | Chloroform anhydrous 99+% | Sigma-Aldrich | Cat# 288306 | |
| Commercial assay or kit | TRIzol Reagent | Thermo Fisher Scientific | Cat# 15596026 | |
| Commercial assay or kit | Insulin solution human | Sigma-Aldrich | Cat# I9278 | |
| Commercial assay or kit | Fibronectin from human plasma 0.1% solution | Sigma-Aldrich | Cat# F0895 | |
| Commercial assay or kit | ProLong Diamond Antifade Mountant | Thermo Fisher Scientific | Cat# P36961 | |
| Commercial assay or kit | Fly Extract | *Drosophila* Genomics Resource Center | Cat# 1645670 | |
| Commercial assay or kit | SMARTer RNA Unique Dual Index Kit – 24U | Clontech | Cat# 634451 | |
| Commercial assay or kit | RiboPOOL 10 nM for *Drosophila* | Cambridge Bioscience | | |
| Commercial assay or kit | D-Biotin solution | Thermo Fisher Scientific | Cat# B20656 | |
| Commercial assay or kit | Shields and Sang M3 Insect Medium | Sigma | Cat# S3652 | |
| Commercial assay or kit | Library Quantification Kit | Kapa Biosystems | Cat# KK4873 | |
| Commercial assay or kit | ProTEV Plus | Promega | Cat# V6101 | |
| Commercial assay or kit | Paraformaldehyde, 16% w/v aq. soln., methanol free | Alfa Aesar | Cat# 043368.9M | |
| Commercial assay or kit | RNase A (affinity purified) 1 mg/ml | Thermo Fisher Scientific | Cat# AM2271 | |
| Commercial assay or kit | NuPAGE 4–12%, Bis-Tris, 1.5 mm, Mini Protein Gel, 10-well | Thermo Fisher Scientific | Cat# NP0335BOX | |

*Continued on next page*

*Continued*

| Reagent type (species) or resource | Designation | Source or reference | Identifiers | Additional information |
|---|---|---|---|---|
| Commercial assay or kit | Xfect Transfection Reagent | Takara Bio | Cat# 631318 | |
| Commercial assay or kit | Pierce Protein A/G Magnetic Beads | Thermo Fisher Scientific | Cat# 88802 | |
| Commercial assay or kit | Agencourt RNAClean XP beads | Beckman Coulter | Cat# A63987 | |
| Commercial assay or kit | NEBNext Ultra Directional RNA Library Prep Kit for Illumina | New England Biolabs | Cat# E7420L | |
| Commercial assay or kit | SMARTer Stranded RNA-Seq Kit | Takara Bio | Cat# 634839 | |
| Commercial assay or kit | RNeasy Mini Kit | Qiagen | Cat# 74104 | |
| Commercial assay or kit | Cell Line Nucleofector kit V | Lonza | Cat# VVCA-1003 | |
| Commercial assay or kit | Nucleofector II device | Lonza | Cat# AAB-1001 | |
| Cell line (*Drosophila melanogaster*) | S2 cells | Thermo Fisher Scientific | Cat# R69007, RRID:CVCL_Z232 | |
| Cell line (*Drosophila melanogaster*) | Ovarian somatic cells (OSCs) | DOI:10.1038/nature08501 | RRID:CVCL_IY73 | |
| Software, algorithm | Fiji | ImageJ | RRID:SCR_002285 | |
| Software, algorithm | Proteome Discoverer 2.1 | Thermo Fisher Scientific | RRID:SCR_014477 | |
| Software, algorithm | STAR | DOI:10.1093/bioinformatics/bts635 | RRID:SCR_015899 | |
| Software, algorithm | DEseq2 | DOI:10.1186/s13059-014-0550-8 | RRID:SCR_015687 | |
| Software, algorithm | Image Studio Lite | LI-COR | RRID:SCR_013715 | |

## Cell lines

OSCs were a gift from Mikiko Siomi and were cultured as described (*Niki et al., 2006*; *Saito, 2014*; *Saito et al., 2009*). *Drosophila* S2 cells were purchased from Thermo Fisher Scientific and were grown at 26°C in Schneider media supplemented with 10% FBS. Cells were routinely tested for mycoplasma infection by an in-house facility.

## Fly stocks and handling

All flies were kept at 25°C on standard cornmeal or propionic food. Control *w1118* flies were a gift from the University of Cambridge Department of Genetics Fly Facility. *Nup54* mutant lines were provided by M. Soller (*Nallasivan et al., 2020*). A full list of fly stocks used in this study is provided in *Supplementary file 1*.

## Cell culture

Knockdowns and nucleofections in OSCs were carried out as previously described (*Saito, 2014*) using the Cell Line Nucleofector Kit V (Lonza VVCA-1003) on a Nucleofector II device (program T-029). OSC transfections were carried out using Xfect transfection reagent (Takara Bio 631317), as previously described (*Saito, 2014*). All constructs used in cells were expressed from the *Drosophila act5c* promoter. A full list of siRNAs used in this study is provided in *Supplementary file 1*. S2 cells were transfected using Effectene (Qiagen), according to the manufacturer's instructions.

## BASU proximity labelling and mass spectrometry

PL-MS experiments in OSCs were performed as previously described (*Munafò et al., 2019*). Briefly, $4 \times 10^6$ OSCs were transfected with 20 μg of plasmid expressing an HA-BASU fusion or HA-ZsGreen. 48 hr after transfection, the media was supplemented with 200 μM biotin for 1 hr. Cell

pellets were lysed in 1.8 ml lysis buffer (50 mM Tris, pH 7.4, 500 mM NaCl, 0.4% SDS, 1 mM DTT, 2% TritonX-100 with cOmplete protease inhibitors) and sonicated using a Bioruptor Pico (Diagenode, 3× cycles 30 s on/30 s off). Sonicated lysates were diluted 2× in 50 mM Tris, pH 7.4, and cleared for 10 min at 16,500 g. Lysates were pre-cleared for 1 hr at 4°C with 100 µl of Protein A/G Dynabeads (Thermo Fisher Scientific 10015D) and the supernatant collected to a fresh tube. Biotinylated proteins were isolated by incubation with 200 µl of Dynabeads (MyOne Streptavidin C1; Life Technologies) overnight at 4°C. Beads were washed 2× in 2% SDS, 2× in Wash Buffer 1 (0.1% deoxycholate, 1% Triton X-100, 500 mM NaCl, 1 mM EDTA, and 50 mM 4-(2-hydroxyethyl)−1-piperazineethanesulfonic acid, pH 7.5), 2× with Wash Buffer 2 (250 mM LiCl, 0.5% NP-40, 0.5% deoxycholate, 1 mM EDTA, and 10 mM Tris, pH 8), and 2× with 50 mM Tris. Beads were rinsed twice with 100 mM ammonium bicarbonate. BASU-Nup54, BASU-Nup58, and BASU-Yb pulldowns were subjected to TMT-labelling followed by quantitative mass spectrometry on a nano-ESI Fusion Lumos mass spectrometer (Thermo Fisher Scientific). On-bead Trypsin digestion, TMT chemical isobaric labelling and data analysis were performed by the CRUK-CI proteomics core as previously described (*Papachristou et al., 2018*).

## Proteomics data analysis

Spectral raw files from PL-MS of BASU-Yb, Nup54, and Nup58 were processed with the SequestHT search engine on Thermo Scientific Proteome Discoverer 2.1. Data was searched against a custom FlyBase database ('dmel-all-translation-r6.24') at 1% spectrum-level FDR (False Discovery Rate) criteria using Percolator (University of Washington). MS1 mass tolerance was constrained to 20 ppm, and the fragment ion mass tolerance was set to 0.5 Da. TMT tags on lysine residues and peptide N termini (+229.163 Da) were set as static modifications. Oxidation of methionine residues (+15.995 Da), deamidation (+0.984) of asparagine and glutamine residues, and biotinylation of lysines and protein N-terminus (+226.078) were included as dynamic modifications. For TMT-based reporter ion quantitation, we extracted the signal-to-noise ratio for each TMT channel. Parsimony principle was applied for protein grouping, and the level of confidence for peptide identifications was estimated using the Percolator node with decoy database search. Strict FDR was set at q-value <0.01. Downstream data analysis was performed on R using the qPLEXanalyzer package (https://doi.org/10.5281/zenodo.1237825) as described (*Papachristou et al., 2018*). Only proteins with more than one unique peptide were plotted.

## Co-immunoprecipitation from cell lysates

S2 cells or OSCs were transfected with 3xFLAG- and HA-tagged constructs. After 48 hr, cells were lysed in 250 µl of Pierce IP lysis buffer supplemented with cOmplete protease inhibitors (Roche). Equal amounts of lysate for each sample were diluted to 1 ml with IP lysis buffer and incubated with 30 µl of anti-FLAG M2 magnetic beads (Sigma M8823) overnight at 4°C. For anti-HA pulldowns, lysates were incubated with 30 µl of anti-HA (Thermo Fisher Scientific 88836) beads overnight at 4°C. Beads were washed 3 × 15 min in 1× Tris-buffered Saline (TBS) with protease inhibitors, then resuspended in 2xNuPAGE LDS Sample Buffer (Thermo Fisher Scientific) and boiled for 3 min at 90°C to elute immunoprecipitated proteins.

## Western blot

Western blots were carried out using standard protocols. Protein lysates were run on NuPAGE 4–12% pre-cast gels and transferred to nitrocellulose membranes using a dry blotting system (iBlot2.0). Membranes were blocked for 1 hr at RT with 1× LiCor blocking buffer diluted in Phosphate Buffered Saline (PBS) and primary antibodies incubated overnight at 4°C. Following 3× washes in PBS + 0.1% Tween, secondary antibodies conjugated to infrared dyes (and/or streptavidin; LiCor925-32230) were incubated for 45 min at room temperature (RT). Images were acquired on an Odyssey CLx scanner (LiCor). The following primary antibodies were used: anti-HA (C29F4; Cell Signaling Technology), anti-FLAG (Sigma #F1804), anti-Piwi (*Brennecke et al., 2007*), anti-Yb (*Saito et al., 2010*), anti-Yb (*Handler et al., 2011*) (used on ovary lysates), and anti-tubulin (ab18251).

## OSCs immunostaining

Cells were plated 1 day in advance on fibronectin-coated coverslips, fixed for 15 min in 4% Paraformaldehyde (PFA), permeabilised for 10 min in PBS, 0.2% Triton X-100, and blocked for 30 min in PBS, 0.1% Tween-20 (PBST), and 1% BSA. Primary antibodies were diluted 1:500 in PBST and 0.1%BSA) and incubated overnight at 4°C. After 3 × 5 min washes in PBST, secondary antibodies were incubated for 1 hr at RT. After 3 × 5 min washes in PBST, DAPI was incubated for 10 min at RT and washed twice in PBST. Coverslips were mounted with ProLong Diamond Antifade Mountant (Thermo Fisher Scientific #P36961) and imaged on a Leica SP8 confocal microscope (100× oil objective). For TurboID labelling, cell culture media was supplemented with 500 µM biotin for 1 hr. Detection was carried out using streptavidin conjugated to AlexaFluor-555 (Thermo Fisher Scientific). The following antibodies were used: anti-Piwi (*Brennecke et al., 2007*), anti-FLAG tag (Sigma #F1804), anti-HA tag (ab9110), anti-Yb (*Saito et al., 2010*), anti-lamin (DSHB, ADL67.10), and anti-FG-Nups (Biolegend mAb414).

## RNA-FISH in OSCs

RNA-FISH was performed with hybridisation chain reaction (HCR), similar as reported (*Ang and Yung, 2016*; *Choi et al., 2014*). OSCs were seeded on fibronectin-coated coverslips, fixed for 15 min in 4% PFA, washed 2 × 5 min with PBS, and permeabilised for at least 24 hr in 70% ethanol at −20°C. Ethanol was removed and slides were washed 2 × 5 min in 2× saline-sodium citrate buffer (SSC). Samples were incubated for 10 min in 15% formamide in 2× SSC. HCR probes were diluted to 1 nM each in hybridisation buffer (15% formamide, 10% dextran sulfate in 2× SSC) and incubated overnight at 37°C in a humidified chamber. Samples were washed twice in 2× SSC and 10 min in 30% formamide at 37°C. HCR hairpins conjugated to AlexaFluor-647 or oligo-dT probes conjugated to AlexaFluor-488 were heat-denatured and diluted to 120 nM in 5× SSC and 0.1% Tween-20 (SSCT). HCR amplification was carried out for 2 hr at RT in the dark and washed 3 × 10 min with 5× SSCT. Nuclei were stained with DAPI (1:10,000 in SSCT) for 10 min, followed by 3 × 10 min washes in 5× SSC. Slides were mounted with ProLong Diamond Antifade Mountant (Thermo Fisher Scientific) and imaged on a Leica SP8 confocal microscope (100× oil objective). Probes were purchased from IDT, and all sequences are provided in *Supplementary file 1*.

## RNA isolation and qPCR analysis

Ovary samples or cell pellets were lysed in 1 ml TRIzol, and RNA was extracted according to the manufacturer's instruction. 1 µg of total RNA was treated with DNAseI (Thermo Fisher Scientific), and reverse transcribed with the Superscript III First Strand Synthesis Kit (Thermo Fisher Scientific), using oligo-dT$_{20}$ primers. Real-time PCR (qPCR) experiments were performed with a QuantStudio Real-Time PCR Light Cycler (Thermo Fisher Scientific). Transposon levels were quantified using the ΔΔCT method (*Livak and Schmittgen, 2001*), normalised to *rp49,* and fold-changes were calculated relative to the indicated controls. All primer sequences are listed in *Supplementary file 1*.

## Ovary immunostaining

Fly ovaries were dissected in ice-cold PBS, fixed for 15 min in 4% PFA at RT, and permeabilised with 3 × 10 min washes in PBS with 0.3% Triton X-100 (PBS-Tr). Samples were blocked in PBS-Tr with 1% BSA for 2 hr at RT and incubated overnight at 4°C with primary antibodies in PBS-Tr and 1% BSA. After 3 × 10 min washes at RT in PBS-Tr, secondary antibodies were incubated overnight at 4°C in PBS-Tr and 1% BSA. After 4 × 10 min washes in PBS-Tr at RT (DAPI was added during the third wash) and 2 × 5 min washes in PBS, samples were mounted with ProLong Diamond Antifade Mountant (Thermo Fisher Scientific #P36961) and imaged on a Leica SP8 confocal microscope. Images were deconvoluted using Huygens Professional. The following antibodies were used: anti-Piwi (*Brennecke et al., 2007*), anti-Yb (*Saito et al., 2010*), and anti-FG Nups (Biolegend mAb414).

## Image analysis on Fiji

Acquired images were analysed on Fiji using custom scripts (*Source code 1* and *2*). Briefly, for Yb-body area measurements we extracted the relative channel, applied a threshold, and analysed particle number and size. A similar number of images was processed for all samples. For *flam* RNA-FISH analysis, we identified nuclei from the lamin staining applying a difference of Gaussian filter. We

then isolated the RNA-FISH spots and counted the number present inside the nuclear envelope versus the total amount. Cytoplasm was identified via oligo-dT staining. A similar number of images was processed for all samples.

## RNA-seq

Ribosomal RNAs were depleted using riboPOOLs against *D. melanogaster* rRNAs (siTOOLs Biotech), according to the manufacturer's instructions. Fly riboPOOLs were hybridised to 1 μg of RNA by adding 1 μl of resuspended riboPOOLs (100 μM), 5 μl of hybridisation buffer (10 mM Tris-HCl pH 7.5, 1 mM EDTA, 2 M NaCl), and 1 μl of RNAse Inhibitor Plus (Promega), incubating for 10 min at 68°C and cooling down slowly to 37°C. 80 μl of MyOne Streptavidin C1 beads (Thermo Fisher) for each sample were washed twice in 100 μl of Beads Resuspension Buffer (0.1 M NaOH, 0.05 M NaCl) and twice in 100 μl of Beads Wash Buffer (0.1 M NaCl). The beads were resuspended in 160 μl of Depletion buffer (5 mM Tris-HCl pH 7.5, 0.5 mM EDTA, 1 M NaCl) and divided into two 80 μl aliquots. Hybridised riboPOOLs were added to 80 μl of washed beads, mixed well, and incubated for 15 min at 37°C, followed by 5 min at 50°C. The supernatant was transferred to the second tube with 80 μl of washed beads and incubated for 15 min at 37°C, followed by 5 min at 50°C. rRNA-depleted samples were transferred to a fresh tube and purified using Agencourt RNAClean XP beads (Beckman Coulter A63987). RNA-seq libraries were prepared using the NEBNext Ultra Directional Library Prep Kit for Illumina (NEB #E7760), according to the manufacturer's instructions for ribosome-depleted RNA. DNA libraries were quantified with KAPA Library Quantification Kit for Illumina (Kapa Biosystems) and sequenced on an Illumina HiSeq 4000 instrument.

## sRNA-seq library preparation

Small RNA libraries from OSCs were generated as described previously with slight modifications (*McGinn and Czech, 2014*). Briefly, 19- to 28-nt small RNAs were purified by PAGE from 15 μg of total RNA from OSCs. Next, the 3′ adapter (containing four random nucleotides at the 5′ end) was ligated using T4 RNA ligase 2, truncated KQ (NEB). Following recovery of the products by PAGE purification, the 5′ adapter (containing four random nucleotides at the 3′ end) was ligated to the samples using T4 RNA ligase (Ambion). Small RNAs containing both adapters were recovered by PAGE purification, reverse transcribed, and PCR amplified. Libraries were sequenced on an Illumina HiSeq 4000 instrument.

## PRO-seq

PRO-seq was performed as described previously (*Mahat et al., 2016*). $4 \times 10^6$ OSCs treated with siRNAs for 96 hr were used for nuclei isolation. Isolated nuclei were resuspended in storage buffer and stored at $-80°C$ until further processing. Nuclear run-on reactions were performed with biotin-11-CTP and biotin-11-UTP and unlabelled ATP and GTP. Purified RNA samples were fragmented for 10 min on ice in 0.2 N NaOH and purified using Bio-Spin P30 columns (Bio-Rad). Biotin-labelled RNA was purified using MyOne Streptavidin C1 Dynabeads (Thermo Fisher Scientific), decapped using the RppH enzyme (NEB), and purified by phenol/chloroform extraction. 3′ linkers and then 5′ linkers (same as for small RNA cloning) were ligated and biotinylated ligation products purified after each ligation using MyOne Streptavidin C1 Dynabeads. Reverse transcription was carried out using SuperScript III (Thermo Fisher Scientific), and libraries were amplified using HS Phusion Flex polymerase. The libraries were sequenced on an Illumina HiSeq 4000.

## DNA-FISH

DNA-FISH was carried out as described in *Kishi et al., 2019*. DNA-FISH probes were designed against a 10 kb region spanning the *DIP1* and *flamenco* genomic loci (chrX:21624796–21634619) using Oligominer (*Beliveau et al., 2018*). The final set of 74 probes were completed with the addition of the SABER primer sequences at their 3′ ends (tttCAACTTAAC) and purchased from IDT. PER amplification was carried out as described by Kishi and colleagues (*Kishi et al., 2019*) and used directly for DNA-FISH. OSCs were plated on fibronectin-coated slides and fixed for 10 min in 4% PFA, permeabilised for 10 min in PBS, 0.5% TritonX-100, and washed twice in PBS with 0.1% Tween-20 (PBST). If necessary, DNase treatment was carried out at this stage by incubation with 4 μl of Turbo DNase in 100 μl of 1× Turbo DNase buffer for 30 min at 37°C. Cells were incubated 5 min in

0.1 N HCl, washed twice in PBST, and incubated in 2× SSCT (2× SSC with 0.1% Tween-20) with 50% formamide for 2 hr at 60°C. Cells were hybridised in 80 µl of ISH solution consisting of 2× SSCT, 50% formamide, 10% dextran sulfate, 400 ng/µl RNase A, and each PER extension at a final concentration of ~67 nM (1:15 dilution from 1 µM PER). After denaturation for 3 min at 80°C, cells were incubated overnight at 44°C in a humidified incubator. Hybridised samples were washed 4 × 5 min in prewarmed 2× SSCT at 60°C and then twice at RT. 80 µl of fluorescent hybridisation solution consisting of 1× PBS and 1 µM fluorescent imager strands were added to the samples and incubated for 1 hr at 37°C. Cells were washed 3 × 5 min in prewarmed PBS at 37°C, stained for 10 min at RT with DAPI (1:1000 dilution in PBS), and mounted using ProLong Diamond Antifade Mountant (Thermo Fisher Scientific #P36961). Samples were imaged on a Leica SP8 confocal microscope (100× oil objective).

## CLIP-seq

$1 \times 10^7$ OSCs were nucleofected with 5 µg of the desired plasmid (HALO-tagged Nup54, Nup58, Nxt1, or a HALO-MCS control) and crosslinked on ice with 400 mJ/cm$^2$ at 254 nm. HALO-tag CLIP-seq was performed as previously described (*Munafò et al., 2019*). Briefly, cell pellets were lysed in 300 µl of lLysis buffer (50 mM Tris-HCl pH 7.5, 150 mM NaCl, 1% Triton X-100, 0.1% deoxycholate, Protease Inhibitor [1:50 Promega], and RNasin Plus [1:500, Promega]), for 30 min at 4°C. DNase digestion was performed by adding 2 µl (4U) of Turbo DNase to the cell lysate and immediately placing the samples at 37°C for 3 min, shaking at 1100 rpm. Samples were transferred to and kept on ice for >3 min, then cleared by centrifugation at top speed for 20 min at 4°C. Cell lysates were diluted up to 1 ml with 100 mM Tris-HCl pH 7.5, 150 mM NaCl, and incubated with 200 µl of Magne-HaloTag (Promega G7282) beads overnight at 4°C. Beads were washed 2× in Wash Buffer A (100 mM Tris-HCl pH 7.5, 150 mM NaCl, 0.05% IGEPAL CA-630), 3× in Wash Buffer B (PBS, 500 mM NaCl, 0.1% Triton X-100, RNasin Plus 1:2000), 3× in PBS, 0.1% Triton X-100, and rinsed in Wash Buffer A. For release of the bait protein from the tag, beads were resuspended in 100 µl of 1X Pro-TEV Buffer, 1 mM DTT, and RNasin Plus (1:50) and 25 units of ProTEV Plus Protease (Promega V6101) and incubated 2 hr at 30°C, shaking at 1300 rpm. The supernatant containing the eluted protein and the crosslinked RNA was transferred to a fresh tube, and 15 µl Proteinase K in 300 µl PK/SDS buffer (100 mM Tris, pH 7.5; 50 mM NaCl; 1 mM EDTA; 0.2% SDS) were added to the eluate and incubated 1 hr at 50°C. RNA was isolated via phenol/chloroform extraction, resuspended in 8 µl of nuclease-free water, and used for library preparation. Library preparation for CLIP-seq samples was carried out with the SMARTer Stranded RNAseq kit (Takara Bio 634839), according to the manufacturer's instructions.

## Sequencing data analysis

For small RNA-seq, adapters were clipped from raw fastq files with fastx_clipper (adapter sequence AGATCGGAAGAGCACACGTCTGAACTCCAGTCA) keeping only reads with at least 23 bp length. The first and last four bases were trimmed using seqtk (https://github.com/lh3/seqtk; *Ramírez et al., 2016*). After removal of cloning markers and reads mapping to rRNAs and tRNAs (list downloaded from FlyBase), high-quality reads were aligned to the *D. melanogaster* genome release 6 (dm6; downloaded from FlyBase) (*Hoskins et al., 2015*) using STAR (*Dobin et al., 2013*). For transposon-wide analysis, genome multi-mapping reads were randomly assigned to one location using option '--outFilterMultimapNmax 1000 --outSAMmultNmax 1 --outMultimapperOrder Random' and non-mapping reads were removed. Small RNA-seq reads were normalised to miRNA reads in the control library (set to rpm). Only high-quality small RNA reads with a length between 23 and 29 bp were used for the piRNA genome browser shots. piRNA distribution was calculated and plotted in R.

For RNA-seq, raw fastq files generated by Illumina sequencing were analysed by a pipeline developed in-house. In short, the first five bases of each 50 bp read were removed using fastx trimmer (http://hannonlab.cshl.edu/fastx_toolkit/). After removal of reads mapping to *Drosophila* rRNA using STAR, high-quality reads were aligned to the *D. melanogaster* genome release 6 (dm6; downloaded from FlyBase) (*Hoskins et al., 2015*) using STAR (*Dobin et al., 2013*). For transposon-wide analysis, genome multi-mapping reads were randomly assigned to one location using option '--outFilterMultimapNmax 1000 --outSAMmultNmax 1 --outMultimapperOrder Random' and non-

mapping reads were removed. For genome-wide analyses, multi-mapping reads were removed to ensure unique locations of reads. Normalisation was achieved by calculating rpm using the deepTools2 (*Ramírez et al., 2016*) bamCoverage. Differential expression analysis was performed using DESeq2 (*Love et al., 2014*) and plotting was done in R using ggplot2 (https://ggplot2.tidy-verse.org). For genes carrying 'nearby transposon insertions', we considered only gene promoters within $-10$ kb to $+15$ kb of a TE insertion on the same genomic strand. Coordinates used for the uni-strand piRNA clusters are chrX:21631891–22282863 (*flam*) and chrX:21520428–21556793 (*20A*). Coordinates used for the dual-strand piRNA clusters were from *Andersen et al., 2017*.

For piRNA clusters 1 kb bin analysis, each cluster was divided into non-overlapping 1 kb bins and only those with a mappability score above 0.8 were retained. Uniquely mapped reads were counted using HTSeq (*Anders et al., 2015*), normalised to the total number of genome-mapped reads, and the per-window $\log_2$ fold-change between each knockdown and its control was calculated and plotted in R. The mappability was calculated as described in *Derrien et al., 2012*. For OSC RNA-seq, bins with 0 rpm in more than one sample were discarded. For ovary RNA-seq analysis, a pseudo-count of 0.01 was added to each bin and the bins with 0 rpm only in the control were discarded. For *flam* 100 kb bins analysis, the dm6 genome was divided into 100 kb sliding windows using 1 kb steps. Mappability for a window was defined as the fraction of all possible 50-mers derived from the window that aligned uniquely to it using STAR. Windows with mappability >0.05 and located fully within *flam* (n = 285) were kept for the analysis. For each sample, reads aligning uniquely to the sense strand and at least 50% within a window were counted and subsequently normalised to reads per 1 million uniquely aligned reads. The per-window $\log_2$ fold-change between each knockdown and its control was calculated using a pseudo-count of 1. Results from four individual replicates per knockdown were highly consistent with the results shown from the pooled analysis.

## Quantification and statistical analysis

Data visualisation and analyses were done using R and the following packages: ggplot2, DEseq2, qPLEXanalyzer. The UCSC genome browser was used to display high-throughput sequencing data and to prepare coverage plots shown in the article. n indicated in the figure legends refers to the number of independent biological replicates. Bar graphs display average of n biological replicates and standard deviation (SD), and p values were calculated with an unpaired *t*-test using GraphPad. Box plots display median, first, and third quartiles (box) and highest/lowest value within 1.5 inter-quartile range (whiskers); dots represent potential outliers beyond 1.5 * interquartile range.

## Acknowledgements

We thank Vera Manelli and Federica A Falconio for help with cloning. We thank the Cancer Research UK Cambridge Institute Bioinformatics, Genomics, Microscopy, RICS, and Proteomics Core Facilities for support, particularly Kamal Kishore and Fadwa Joud. We thank the Life Science Editors, especially Marie Bao, and members of the Hannon lab for feedback and comments on the manuscript. We thank the University of Cambridge Department of Genetics Fly Facility for microinjection services and fly stock generation. We thank the Vienna *Drosophila* Resource Center and the Bloomington Stock Center for fly stocks. We thank Mikiko Siomi for OSCs and anti-Yb antibody, and Julius Brennecke for anti-Yb antibodies. MM was supported by a Boehringer Ingelheim Fonds PhD fellowship. MS acknowledges funding from the BBSRC. GJH is a Royal Society Wolfson Research Professor (RP130039). Research in the Hannon laboratory is supported by Cancer Research UK and a Wellcome Trust Investigator award (110161/Z/15/Z).

## Additional information

### Funding

| Funder | Grant reference number | Author |
| --- | --- | --- |
| Cancer Research UK | Core funding (A21143) | Gregory J Hannon |
| Wellcome Trust | Investigator award (110161/Z/15/Z) | Gregory J Hannon |

| Royal Society | Wolfson Research Professor (RP130039) | Gregory J Hannon |
|---|---|---|
| Boehringer Ingelheim Fonds | PhD fellowship | Marzia Munafò |
| Biotechnology and Biological Sciences Research Council | | Matthias Soller |

The funders had no role in study design, data collection and interpretation, or the decision to submit the work for publication.

### Author contributions
Marzia Munafò, Conceptualization, Data curation, Software, Formal analysis, Supervision, Funding acquisition, Validation, Investigation, Visualization, Methodology, Writing - original draft, Writing - review and editing; Victoria R Lawless, Alessandro Passera, Serena MacMillan, Validation, Investigation, Writing - review and editing; Susanne Bornelöv, Data curation, Software, Formal analysis, Validation, Visualization, Writing - review and editing; Irmgard U Haussmann, Resources, Writing - review and editing; Matthias Soller, Resources, Funding acquisition, Writing - review and editing; Gregory J Hannon, Conceptualization, Supervision, Funding acquisition, Methodology, Project administration, Writing - review and editing; Benjamin Czech, Conceptualization, Data curation, Formal analysis, Supervision, Validation, Investigation, Visualization, Methodology, Writing - original draft, Project administration, Writing - review and editing

### Author ORCIDs
Marzia Munafò ![ORCID] https://orcid.org/0000-0002-2689-8432
Victoria R Lawless ![ORCID] http://orcid.org/0000-0003-0406-6552
Susanne Bornelöv ![ORCID] http://orcid.org/0000-0001-9276-9981
Irmgard U Haussmann ![ORCID] http://orcid.org/0000-0002-2764-694X
Matthias Soller ![ORCID] http://orcid.org/0000-0003-3844-0258
Gregory J Hannon ![ORCID] https://orcid.org/0000-0003-4021-3898
Benjamin Czech ![ORCID] https://orcid.org/0000-0001-8471-0007

### Decision letter and Author response
Decision letter https://doi.org/10.7554/eLife.66321.sa1
Author response https://doi.org/10.7554/eLife.66321.sa2

## Additional files
### Supplementary files
- Source code 1. Custom script for Yb-bodies analysis in Fiji.
- Source code 2. Custom script for *flam* RNA-FISH analysis in Fiji.
- Supplementary file 1. List of siRNA sequences, qPCR primers, and fly stocks used in this study.
- Transparent reporting form

### Data availability
Sequencing data have been deposited in GEO under accession number GSE152297. Mass spectrometry data have been deposited to the PRIDE Archive under accessions PXD019670 and PXD019671. Source data files have been provided for Figure 4.

The following datasets were generated:

| Author(s) | Year | Dataset title | Dataset URL | Database and Identifier |
|---|---|---|---|---|
| Munafò M, Lawless RV, Passera A, MacMillan S, Bornelöv S, Haussmann IU, | 2021 | Channel Nuclear Pore Complex subunits are required for transposon silencing in *Drosophila* | https://www.ncbi.nlm.nih.gov/geo/query/acc.cgi?acc=GSE152297 | NCBI Gene Expression Omnibus, GSE152297 |

| | | | | | |
|---|---|---|---|---|---|
| Soller M, Hannon GJ, Czech B | | | | | |
| Munafò M, Lawless RV, Passera A, MacMillan S, Bornelöv S, Haussmann IU, Soller M, Hannon GJ, Czech B | 2021 | Channel Nuclear Pore Complex subunits are required for transposon silencing in *Drosophila* | https://www.ebi.ac.uk/pride/archive/projects/PXD019671 | | PRIDE, PXD019671 |
| Munafò M, Lawless RV, Passera A, MacMillan S, Bornelöv S, Haussmann IU, Soller M, Hannon GJ, Czech B | 2021 | Channel Nuclear Pore Complex subunits are required for transposon silencing in *Drosophila* | https://www.ebi.ac.uk/pride/archive/projects/PXD019670 | | PRIDE, PXD019670 |

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
