## [Decision Letter]

**Acceptance summary:**

A central unsolved problem in the piRNA field is, what marks an RNA for piRNA production? Here, Czech and co-workers report that the nucleoporins Nup54 and Nup58 play a specialized role in specifying the flamenco transcript for processing into piRNAs, an essential step for silencing gypsy-family transposons in the somatic follicle cells of the fly ovary. Although it is not clear whether these findings will inform our understanding of piRNA biogenesis beyond the *melanogaster* subgroup of flies, the work is nonetheless important to the piRNA field generally, since the mechanisms that distinguish long non-coding RNAs from piRNA precursor transcripts are not understood in any animal.

**Decision letter after peer review:**

Thank you for submitting your article "Channel Nuclear Pore Complex subunits are required for transposon silencing in *Drosophila*" for consideration by *eLife*. Your article has been reviewed by 3 peer reviewers, and the evaluation has been overseen by a Reviewing Editor and Kevin Struhl as the Senior Editor. The reviewers have opted to remain anonymous.

Essential revisions:

The three reviewers agree that this is overall a compelling study that provides novel, interesting and important observations. No major new experimentation is required but the data need to be more extensively described and the authors should also discuss alternative models. Detailed suggestions for changes are found in the constructive reviews but it is important that the authors improve the quality of some of their images, provide additional quantitative information and add better statistical analyses.

*Reviewer #1:*

The authors set out to identify nuclear pore components required for piRNA production in cultured *Drosophila melanogaster* OSC cells, a model for the somatic follicle cells that support oogenesis in the female fly. Detailed measurements of RNA production, piRNA production, and protein-protein interactions are a major strength of this work. In general, experiments using OSC cells are informative and conservatively interpreted. One weakness is the low quality confocal images (perhaps they were uploaded at too low a resolution?) make it difficult to evaluate the authors' descriptions of the location and abundance of specific subcellular components in cells. Only a small number (often just one) of representative images are shown, making it impossible to evaluate how representative any image is of the phenotype of RNAi depletion of a specific nucleoporin or other protein. Most importantly, the authors often ignore alternative explanations more consonant with the nucleopore literature. Interpretation of their genetic experiments is complicated by changes in the cell composition of mutant ovaries. The work will provide a starting point for the dissection of the flam RNA export pathway and the commitment of flam transcripts to piRNA biogenesis.

1. Please include a panel of representative images for mutant phenotypes in OSC and ovaries, not simply a single cell or field of view.

2. Figure 2A: what is the source of the sense piRNAs whose abundance is unaltered by loss of Yb, Nup54, or Nup58? Do they map to cluster transcripts or elsewhere in the genome?

3. The authors propose that Nup54/58 play a role in export of flam RNA. Given the bias towards loss of 3′ transcript coverage and piRNA abundance, together with both the involvement of the TREX complex in piRNA biogenesis and the role of nuclear pore proteins in transcriptional elongation, isn't it more likely that Nup54/58 promote transcriptional elongation across the entirety of flam or that they prevent premature termination in the 3′ region of the flam gene? The authors should at least discuss these alternative explanations, detailing the evidence supporting or refuting each.

4. Page 18: since the mutant ovaries are small (please show ovary and ovariole images of what exactly this means!), then the cell composition of the controls and mutants will be quite different. Early stage ovaries will have a higher the ratio of follicle cells cytoplasm to nurse cells, since nurse cells will be smaller. Thus, it is not simple to normalize RNA or piRNA sequencing data to permit comparison between the two types of ovaries. This problem is exacerbated by the authors' finding that both TE and mRNA expression changes in nup54MB/9B4. A detailed description of the experimental evidence that explains why the authors' normalization strategy allows such a comparison is essential. Additionally, the authors need to show (e.g., by in situ hybridization) that TE expression is in fact higher in vivo in identical cell types.

*Reviewer #2:*

The paper by Munafo et al. addresses the role of specific fly NPC proteins in the export and accumulation of flam transcripts in Yb bodies, condensates of the Yb RNA helicase, associated with the cytoplasmic side of NPCs in oocyte somatic cells (OSCs). Yb bodies are important for the biogenesis/maturation of flam piRNAs involved in the repression of specific transposable elements (TEs such as gipsy, blood, mdg1..) in a still poorly understood process. Yb bodies depend on flam transcription and the Nxf1/Nxt1 nuclear export factors, but it is still unclear how the transcripts are directed to Yb-bodies and from there licensed for processing.

In this paper, the authors show that in OSCs, the loss of two Nups, Nup54 and Nup58, which form a bigger complex with Nup62 and Nup93, specifically interferes with the export of flam RNAs but not other mRNAs. Nup54/58 physically interact with Nxf1/Nxt1 as well as Yb suggesting an export-coupled localization mechanism that specifies flam as a piRNA precursor.

The study uses a wide variety of approaches, including transcriptomics, proteomics, biochemical analyses and imaging in different genetic backgrounds both in vitro using OSCs and in vivo using ovary tissue, to demonstrate the specific role of Nup54 and Nup58 in piRNA biogenesis. The paper presents a very large set of experiments with the following main observations:

1. Knock down of Nup62 and Nup93 results in flam expression defects and upregulation of transposons as well as deregulation of multiple genes. In contrast, knock down of Yb, Nup54 and Nup58 leads to loss or dispersed Yb-bodies, decreased flam RNA levels as well as increased TE expression without substantially affecting overall gene expression or viability. The observations support that Nup54 and Nup58 have a dedicated role in flam RNA export and functional Yb-body formation.

By fluorescence imaging approaches, they also show that Yb formation in OSCs depends on correct flam expression and functional Nxf1/Nxt1 export factors.

2. RNA seq of cells depleted for Nup54 and Nup58 or Yb show reduced flam RNA levels. PROseq data show that these lower levels are due to loss of RNA stability rather than decreased transcription. Instability increases towards the 3'end of the flam locus, suggesting that Nup54/58 are required to ensure processivity of nuclear export of an otherwise unstable transcript. siNup54/58 specifically affect the stability of flam piRNAs while siYb affects the levels of different types of piRNAs indicating a specific role for Nup54/58 in flam piRNA biogenesis/maturation that is distinct from Yb.

3. Loss of Nup54/58 affects flam RNA detection in the cytoplasm, but not in the nucleus where it is detected as a dot corresponding to the flam locus associated with the nuclear periphery. Loss of Nup54/58 also results in decrease and dispersion of Yb bodies. In contract siYb results in reduction of both nuclear and cytoplasmic flam foci supporting the different roles of Yb and Nup54/58 in flam piRNA biogenesis.

Together, RNAseq/Proseq and imaging data support the view that Nup54/58 may ensure processivity of flam nuclear export, which otherwise gets degraded.

4. Deletion mutants and complementation analyses in OSCs indicate that the Nup54/58 C-terminal domains, but not the FG-repeats, are important for efficient TE repression. The C-terminal domain is important for Nup54-Nup58 interaction and Yb body formation.

Accordingly, a natural nup54MB C-terminal mutant shows decreased Yb body formation and increased TE expression in vivo.

5. Finally the authors performed quite sophisticated mass spec analyses to identify factors interacting with either Yb or Nup54/58. Nup54/58 PL-MS identified Yb as well as Nxf1 as interactors. In contrast Yb PL-MS detected Nup214 but failed to detect Nxf1 or Nup54/58, which could possibly reflect the transient interaction of Yb with NPCs.

The mass spec data were then validated by an important series of co-IP experiments between wild-type or mutant forms of the Nup54/58 and Yb proteins to define the interaction domains between these proteins key for flam piRNA biogenesis and TE repression.

Overall, the experiments are well-controlled and the results, presented in multiple graphs with statistical analyses, are extremely convincing. The paper is clearly written and the data are on the whole supportive of the proposed model. This work is well suited for *eLife* and should be acceptable for publication provided the authors address few comments/questions.

General remarks/questions:

The paper is quite dense and sometimes difficult to follow when going back and forth between the multiple figures. It may be worse trying to simplify in order to strengthen the overall message.

Figure 2 figure supplement 1G: this experiment suggests that flam transcription is affected in siYb, but not in sinup54/58. How could loss of Yb affect flam transcription efficiency?

Figure 2 figure supplement 2G: this figure indicates that the Yb bodies and the flam DNA don't show an obvious proximity or correlation (on either side of the NPC). This is unexpected based on the model that Nup54/58 establish a physical connection between Nxf1/Nxt1-bound flam RNAs and their delivery to Yb bodies on the cytoplasmic side of the NPC. This observation is not really discussed in the text and should be clarified since it is somewhat questioning the model presented in Figure 4G.

Figure 2 figure supplement 2H is shown too early since it is linked to Figure 3C.

Figure 3A: the authors should indicate what the red domain in Nup54 corresponds to. Is it the same region lacking in the nup54MB mutant?

Figure 3 figure supplement 1E: Yb staining shows quite clearly that the nup54MD presents reduced Yb body formation. However, it is not clear why DNA staining reveals larger foci in the nup54MD mutant?

Figure 4 and supplements: the mass spec data indicate that Nup54/58 interact with Nxf1 and Yb, consistent with their proposed role in coupling flam RNA export and its addressing to Yb bodies. However, Yb does not efficiently interact with Nup54/58 nor Nxf1 by mass spec. It is unexpected that Yb does not interact with Nxf1, since Nxf1 is expected to transport flam RNAs to the cytoplasmic side of the NPC through interactions with Nup54/Nup58 in order to deliver the transcripts to Yb. co-IP experiments (Figure 4 figure supplement 1D) then show that Yb weakly interacts with both Nup54/Nup58 and Nxt1. Why do the authors think that the weak interaction detected between Yb and Nup54/58 is more functional than the weak interaction between Yb and Nxt1?

When does Yb bind Flam transcripts and wouldn't it be expected that Yb interacts with Nxf1/Nxt1 to take over the flam transcripts? This question should be more clearly addressed in the text or discussion.

*Reviewer #3:*

This work examines the role of two nucleoporins in the export of the flam RNA, which is a precursor for piRNA biogenesis necessary for transposon silencing in *Drosophila* ovaries. There is considerable interest in exploring how nucleoporins function outside of their canonical roles in nuclear transport. Thus, this study is relevant from a fundamental cell biological viewpoint but it also may illuminate how some nups appear to be uniquely required during embryonic development and why nup mutations sometimes result in tissue-specific disease. The paper is impressive in scope and in its use of systems-level, biochemical and cell biological approaches to explore changes to transcription, piRNA biogenesis and biochemical interactions. Further, in general, the data are of high quality although often missing statistical analyses to help evaluate highly variable data. The biggest challenge is that there is so much data crammed into a small number of figures that the paper is very frustrating to navigate. Thus, while it is thoroughly convincing that there is a unique requirement for Nup54 and Nup58 in piRNA biogenesis, specifically from the flam locus, a more deliberate and more detailed description of the data would vastly improve the work. Further, the proposed model may be premature and additional considerations regarding where nups might function in this pathway e.g. in the nucleus or cytoplasm should be more thoroughly considered.

1. There are so many experimental techniques and analyses used in this paper that, when coupled to the often obscure nomenclature of genes and mutant alleles, makes navigating this paper extremely frustrating, particularly for a non-expert. One (of many) example of this is the first figure call out, Figure 1, figure supplement 1, which refers to an 8 panel figure that is essentially not described. The strong suggestion is to expand the text to more fully describe all of the figures to do the work more justice and allow readers access.

2. The strength of the work lies in the remarkable specificity exhibited by loss of function of Nup54/58 to piRNA biogenesis, particularly at the flam locus. However, the model proposed is not fully supported by the data. A key concern is that it is well established that nups can function outside of the NPC, both at genomic loci and also in the cytoplasm but this is not fully considered or ruled out. The suggestion is to be more circumspect with the proposed model and to consider alternative possibilities. The key may lie in the fact that it is the coiled-coil domains of the nups, and not their FG-motifs that appear to be important. Although one could argue that this supports their integration into the NPC, how these domains could connect to Nxf1 without FG-repeats remains difficult to rationalize.

3. Invoking a model that suggests that Yb might act similarly to Dbp5 on the cytosolic filaments is certainly interesting, but the data do not yet support this. A clearer understanding of whether the interaction between Yb and Nup214 is direct would be necessary, further, it should be ruled out that DDX19 is not involved with the export of the flam RNA.

Lastly, in general the data is well quantified but better descriptions of the plots and statistics are really needed for interpretation.

---

## [Author Response]

Essential revisions:The three reviewers agree that this is overall a compelling study that provides novel, interesting and important observations. No major new experimentation is required but the data need to be more extensively described and the authors should also discuss alternative models. Detailed suggestions for changes are found in the constructive reviews but it is important that the authors improve the quality of some of their images, provide additional quantitative information and add better statistical analyses.

We thank the reviewers for their positive evaluation of our manuscript. We have revised the text to expand the introduction, discuss alternative models, and provide more explanation of the data. Furthermore, we have added statistical analysis where relevant and clearly described the analysis in both the figure legends and in the Methods section (see Quantification and Statistical analysis). Please, find below our response to each of the points raised by the reviewers and detailed description of the changes made.

Reviewer #1:[…] 1. Please include a panel of representative images for mutant phenotypes in OSC and ovaries, not simply a single cell or field of view.

We have added additional representative images for OSC and ovary phenotypes and enlarged the panels where possible, see Figure 1—figure supplement 3, Figure 2—figure supplement 3A, Figure 3—figure supplement 2 and Figure 4—figure supplement 1C. Each knockdown/mutant is accompanied by the relative quantification of the size and number of Yb-bodies. We have provided high-resolution files for each figure, which should improve the image quality over the previous version seen by the reviewers (where the figures were embedded in a word document and subsequently exported to PDF, losing quality at each compression). Individual images of each figure panel can be provided upon request.

2. Figure 2A: what is the source of the sense piRNAs whose abundance is unaltered by loss of Yb, Nup54, or Nup58? Do they map to cluster transcripts or elsewhere in the genome?

We have added an additional panel to answer the reviewer’s question (Figure 2—figure supplement 2B). Most sense piRNAs detected upon knockdown of *yb*, *nup54* and *nup58* map to genomic TE insertions, predominantly of the *gypsy* retrotransposon. Since these sense piRNAs are relatively few in the *siGFP* control and their abundance correlates with the degree of TE up-regulation in the various knockdowns, we conclude that they are most likely derived from processing of re-activated transposon transcripts. The same analysis on antisense piRNAs is shown for comparison.

3. The authors propose that Nup54/58 play a role in export of flam RNA. Given the bias towards loss of 3′ transcript coverage and piRNA abundance, together with both the involvement of the TREX complex in piRNA biogenesis and the role of nuclear pore proteins in transcriptional elongation, isn't it more likely that Nup54/58 promote transcriptional elongation across the entirety of flam or that they prevent premature termination in the 3′ region of the flam gene? The authors should at least discuss these alternative explanations, detailing the evidence supporting or refuting each.

We have added discussion of this potential explanation in the text. Overall, we agree with the reviewer that an effect on transcriptional elongation and/or termination is plausible, however our data are not sufficient to strongly support or refute either explanation. We noticed that knockdown of *yb* leads to a mild reduction of PRO-seq signal over the entire *flam* cluster locus, possibly hinting towards a negative effect on transcriptional elongation in cases where this transcription-coupled export axis is disrupted. In conclusion, we believe that our data support the conservative hypothesis of Nup54 and Nup58 involvement in *flam* processive export, with destabilisation of the transcript upon their absence. Future work will shed light on the mechanisms underlying *flam* transcription, which is beyond the scope of the present study.

4. Page 18: since the mutant ovaries are small (please show ovary and ovariole images of what exactly this means!), then the cell composition of the controls and mutants will be quite different. Early stage ovaries will have a higher the ratio of follicle cells cytoplasm to nurse cells, since nurse cells will be smaller. Thus, it is not simple to normalize RNA or piRNA sequencing data to permit comparison between the two types of ovaries. This problem is exacerbated by the authors' finding that both TE and mRNA expression changes in nup54MB/9B4. A detailed description of the experimental evidence that explains why the authors' normalization strategy allows such a comparison is essential. Additionally, the authors need to show (e.g., by in situ hybridization) that TE expression is in fact higher in vivo in identical cell types.

We have now provided a brightfield image of ovaries from *nup54^MB^* trans-heterozygote mutants. As one can appreciate, the mutant ovary morphology is grossly normal and only slightly smaller in size compared to controls, unlike other piRNA pathway mutants described in the literature. We did not observe any striking changes in the relative abundance of nurse or follicle cells, and therefore have no reason to question our normalisation strategy, which is routinely used in the field. We would like to emphasise that other published piRNA pathway mutants/knockdowns have far more rudimentary ovaries than our *nup54^MB/9B4^*, e.g. the *piwi* mutant and somatic knockdown of *piwi* in Olivieri et al., 2010. Another example is from our group (Eastwood et al., *eLife*, 2021), where we showed that germline knockdown of *ctp* leads to severe atrophy of the ovaries (Figure 1G). The corresponding RNA-seq normalisation has been carried out similarly to the one we used in this study.

Reviewer #2:[…] General remarks/questions:The paper is quite dense and sometimes difficult to follow when going back and forth between the multiple figures. It may be worse trying to simplify in order to strengthen the overall message.

We thank the reviewer for the feedback, we have now expanded the text to provide more detailed explanations and to clarify the overall message.

Figure 2 figure supplement 1G: this experiment suggests that flam transcription is affected in siYb, but not in sinup54/58. How could loss of Yb affect flam transcription efficiency?

This result, which is in accordance with previous findings (Murota et al., 2014), might be explained in two possible ways. First, residual *flam* RNA that is exported via Nup54/58 does not aggregate into discrete foci, due to absence of Yb, and is therefore hardly detectable by RNA-FISH. Second, our PRO-seq data shows a mild reduction in global *flam* nascent RNA levels upon loss of *yb*, which is more apparent towards the 3’ end of the locus. Although the molecular mechanism is not readily apparent, this might support the idea that *flam* transcriptional elongation is somehow dependent on its downstream processing into piRNAs. A minor fraction of *flam*-derived piRNAs is antisense to the cluster, so we cannot exclude that those might base pair with the nascent *flam* RNA and contribute to its transcriptional rate. Further investigation of the mechanisms underlying *flam* cluster definition and transcription will be required to clarify this.

Figure 2 figure supplement 2G: this figure indicates that the Yb bodies and the flam DNA don't show an obvious proximity or correlation (on either side of the NPC). This is unexpected based on the model that Nup54/58 establish a physical connection between Nxf1/Nxt1-bound flam RNAs and their delivery to Yb bodies on the cytoplasmic side of the NPC. This observation is not really discussed in the text and should be clarified since it is somewhat questioning the model presented in Figure 4G.

This observation is in line with previous works (Dennis et al., 2013; 2016). We suggest that the nucleation of Yb-bodies initiates wherever *flam* transcripts are exported, irrespective of their transcription site.

Figure 2 figure supplement 2H is shown too early since it is linked to Figure 3C.

We thank the reviewer for this slip. Former panel 2H has now been moved to Figure 3—figure supplement 1A.

Figure 3A: the authors should indicate what the red domain in Nup54 corresponds to. Is it the same region lacking in the nup54MB mutant?

The purple box indicates a Nup54-family domain, which is absent in the *nup54^MB^* mutant. This information was already included in the legend of Figure 2A and we have now clarified the description of the *nup54^MB^* allele also in the main text.

Figure 3 figure supplement 1E: Yb staining shows quite clearly that the nup54MD presents reduced Yb body formation. However, it is not clear why DNA staining reveals larger foci in the nup54MD mutant?

The *nup54^MB^* allele is a hypomorph. Some ovarioles of the *nup54^MB^*^/*9B4*^ trans-heterozygotes seem to have a less organised pattern of follicle cells, which often show slightly larger DNA foci. These ovarioles show the most pronounced decrease in Yb-body assembly, thus suggesting that they are the ones where the effects of the Nup54 mutation are most pronounced. However, this phenotype is not fully penetrant and we have now added a second representative image where the DNA foci are more similar to the control.

Figure 4 and supplements: the mass spec data indicate that Nup54/58 interact with Nxf1 and Yb, consistent with their proposed role in coupling flam RNA export and its addressing to Yb bodies. However, Yb does not efficiently interact with Nup54/58 nor Nxf1 by mass spec. It is unexpected that Yb does not interact with Nxf1, since Nxf1 is expected to transport flam RNAs to the cytoplasmic side of the NPC through interactions with Nup54/Nup58 in order to deliver the transcripts to Yb. co-IP experiments (Figure 4 figure supplement 1D) then show that Yb weakly interacts with both Nup54/Nup58 and Nxt1. Why do the authors think that the weak interaction detected between Yb and Nup54/58 is more functional than the weak interaction between Yb and Nxt1?When does Yb bind Flam transcripts and wouldn't it be expected that Yb interacts with Nxf1/Nxt1 to take over the flam transcripts? This question should be more clearly addressed in the text or discussion.

Our co-immunoprecipitation experiments detected a weak interaction between Yb and Nxt1, but not between Yb and Nxf1. Since Nxf1 and Nxt1 are usually present as a heterodimer, we did not speculate further about this interaction being functional. We agree with the reviewer that the “handover” of *flam* transcripts from Nxf1 to Yb is not entirely clear and may involve interactions between Yb and Nxf1/Nxt1 or other adaptor proteins that we have not yet identified. We believe that super-resolution imaging of individual NPCs might clarify the relative position of each component of this export route and shed light onto this point. We have now added a sentence on this to the discussion.

Reviewer #3:This work examines the role of two nucleoporins in the export of the flam RNA, which is a precursor for piRNA biogenesis necessary for transposon silencing in *Drosophila* ovaries. There is considerable interest in exploring how nucleoporins function outside of their canonical roles in nuclear transport. Thus, this study is relevant from a fundamental cell biological viewpoint but it also may illuminate how some nups appear to be uniquely required during embryonic development and why nup mutations sometimes result in tissue-specific disease. The paper is impressive in scope and in its use of systems-level, biochemical and cell biological approaches to explore changes to transcription, piRNA biogenesis and biochemical interactions. Further, in general, the data are of high quality although often missing statistical analyses to help evaluate highly variable data. The biggest challenge is that there is so much data crammed into a small number of figures that the paper is very frustrating to navigate. Thus, while it is thoroughly convincing that there is a unique requirement for Nup54 and Nup58 in piRNA biogenesis, specifically from the flam locus, a more deliberate and more detailed description of the data would vastly improve the work. Further, the proposed model may be premature and additional considerations regarding where nups might function in this pathway e.g. in the nucleus or cytoplasm should be more thoroughly considered.1. There are so many experimental techniques and analyses used in this paper that, when coupled to the often obscure nomenclature of genes and mutant alleles, makes navigating this paper extremely frustrating, particularly for a non-expert. One (of many) example of this is the first figure call out, Figure 1, figure supplement 1, which refers to an 8 panel figure that is essentially not described. The strong suggestion is to expand the text to more fully describe all of the figures to do the work more justice and allow readers access.

We thank the reviewer for this feedback. We have expanded the text to provide more detailed explanations of the data.

2. The strength of the work lies in the remarkable specificity exhibited by loss of function of Nup54/58 to piRNA biogenesis, particularly at the flam locus. However, the model proposed is not fully supported by the data. A key concern is that it is well established that nups can function outside of the NPC, both at genomic loci and also in the cytoplasm but this is not fully considered or ruled out. The suggestion is to be more circumspect with the proposed model and to consider alternative possibilities. The key may lie in the fact that it is the coiled-coil domains of the nups, and not their FG-motifs that appear to be important. Although one could argue that this supports their integration into the NPC, how these domains could connect to Nxf1 without FG-repeats remains difficult to rationalize.

We have adjusted the model description (now referred to as a “tentative model”) and clearly stated what knowledge gaps will need to be filled in order to draw a conclusive picture of *flam* export. We agree that our data do not fully rule out a possible role of Nup54 and Nup58 outside of the NPC, perhaps within Yb-bodies. However, we believe that in such scenario they should have been detectable in our Yb PL-MS and more cytosolic signal would have emerged via immunofluorescence.

3. Invoking a model that suggests that Yb might act similarly to Dbp5 on the cytosolic filaments is certainly interesting, but the data do not yet support this. A clearer understanding of whether the interaction between Yb and Nup214 is direct would be necessary, further, it should be ruled out that DDX19 is not involved with the export of the flam RNA.

We agree with the reviewer’s comment and we have removed the speculation about Yb acting similarly to Dbp5. Further experiments beyond this study are required to address this question.

Lastly, in general the data is well quantified but better descriptions of the plots and statistics are really needed for interpretation.

We have expanded the figure legends to provide better description of the plots and statistics and have added the exact individual p values to each measurement. All quantifications and statistical analyses are accurately described in the methods section.